# Multilayer networks reveal the spatial structure of seed-dispersal interactions across the Great Rift landscapes

Sérgio Timóteo [1], Marta Correia[1], Susana Rodríguez-Echeverría [1], Helena Freitas[1] & Ruben Heleno[1]

Species interaction networks are traditionally explored as discrete entities with well-defined spatial borders, an oversimplification likely impairing their applicability. Using a multilayer network approach, explicitly accounting for inter-habitat connectivity, we investigate the spatial structure of seed–dispersal networks across the Gorongosa National Park, Mozambique. We show that the overall seed–dispersal network is composed by spatially explicit communities of dispersers spanning across habitats, functionally linking the landscape mosaic. Inter-habitat connectivity determines spatial structure, which cannot be accurately described with standard monolayer approaches either splitting or merging habitats. Multilayer modularity cannot be predicted by null models randomizing either interactions within each habitat or those linking habitats; however, as habitat connectivity increases, random processes become more important for overall structure. The importance of dispersers for the overall network structure is captured by multilayer versatility but not by standard metrics. Highly versatile species disperse many plant species across multiple habitats, being critical to landscape functional cohesion.

[1] CFE – Centre for Functional Ecology, Department of Life Sciences, University of Coimbra, Calçada Martim de Freitas, 3000-456 Coimbra, Portugal. Correspondence and requests for materials should be addressed to S.T. (email: stimoteo@gmail.com)

Over the recent decades, ecological networks have proved a valuable framework to simultaneously evaluate the role of species, their interactions, and the importance of the emerging community structure for the persistence and stability of biological communities[1]. Such studies revealed that ignoring the complex web of interactions between plants and animals in which many vital ecosystem functions are rooted might jeopardize the long-term functioning and persistence of ecosystems[2,3]. To date, most studies have considered networks as entities with discrete borders defined by the experimental design, ignoring the potential across-border connections[4], or alternatively as aggregations of several spatially and temporal sampling occasions into an overall network[5]. In nature, however, these sub-networks are linked by common species and by processes that span over several spatial and temporal scales, contributing to the functional connectivity of ecosystems[6]. The importance of the spatial dimension of networks of interactions is becoming increasingly recognized[7–9], highlighting the key function of species that cross habitat boundaries acting as mobile links[10] that connect the different habitats. Recent work has provided further evidence of the importance of the often-neglected inter-habitat links and their unequivocal ecological relevance[11]. Perhaps ironically, the application of such tools that proved particularly suited to tackle the intrinsic complexity of ecosystems is limited by the amount of complexity that can be sampled and analyzed, leading to a fragmentation of real networks, likely to result in oversimplifications, and eventually to incomplete or erroneous conclusions about network structure, dynamics, and stability[12,13]. Similarly, ignoring the role of different species as spatial couplers of ecosystems may hinder our understanding of natural processes, e.g., the flux of energy, or nutrients, between aquatic and terrestrial systems, pollen transfer by insects across the landscape, or the dispersal of seeds of invasive species by birds[14]. Recently, some authors have started to tackle this issue by treating different habitats, or patches of habitat as a set of layers within a larger multilayer network[15–17]. An expansion of the concept of beta-diversity has been proposed to measure dissimilarities between networks, by exploring species and interactions turnover between groups of independent (i.e., formally disconnected) networks[16,18]. In a further step, Frost et al.[15] quantified the connectivity between spatial layers (habitats) of a host-parasitoid network, though they did not explore the effect of habitat connectivity to the structure of the spatial network. However, only now ecologists have started to explicitly include interlayer edges in the analysis of the actual structure of "ecological multilayer networks"[12], taking advantage of recent theoretical developments and analytic tools from other research areas[13,19].

A key structural pattern in networks is modularity[1,20,21], measuring the extent to which species form cohesive groups (modules) where species interact more often within the same module than with species in other modules[22]. These modules provide insights into the phylogenetic history and trait convergence of unrelated species, resulting from local co-adaption, and ecological convergence in the use of resources[23]. By measuring multilayer modularity, the connectivity between layers, i.e., interlayer edge strength, is explicitly accounted for, with the advantages of detecting modules that span across layers. It also allows the identification of nodes that can belong to different modules in different layers, thus particularly relevant for maintaining the continuity of ecosystem functions in space or time[12,24]. However, the ideal way to quantify interlayer edge strength is still a matter of research in multilayer network research, and the investigation of the relative importance of intra- and interlayer processes is essential to understand the structure of multilayer networks[12,24,25].

Contrarily to the high spatial turnover in species and interactions, the functional role of species is considered relatively stable[26]. Centrality measures have been largely used to assess the topological position of a species in the structure of networks[27,28]. In a multilayer context, such overall centrality can be estimated with Google's PageRank[29] algorithm, which has been successfully used[29] to guide conservation strategies[30].

Here, we investigate how a mutualistic multilayer network is structured across habitats and the importance of species to the cohesion of seed dispersal across a complex landscape. To this end, we collected seed–dispersal interactions across the Gorongosa National Park, Mozambique, to build the most complete, seed–dispersal network of the African continent to date[31], including all potential guilds of seed dispersers. Gorongosa underwent a severe defaunation that affected many of the large herbivores, and its recovery is now en route[32,33]. In this context, seed–dispersal is particularly vital for plants to recolonize newly available patches or disturbed ground[34], and is likely a key driver of long-term habitat dynamics in Gorongosa patchy landscapes[35]. Our objectives are twofold. First, we aim to explore the spatial distribution of seed–dispersal modules (i.e., communities of tightly interacting plants and their dispersers) spanning across the different habitats of the Gorongosa National Park. We will do so by evaluating the modularity of multilayer networks formed by discrete, yet interconnected layers representing different habitats. We used different null models to explore how the strength of the interlayer connectivity affects the overall structure of the spatial multilayer network, and to what extent this multilayer approach improves the currently used monolayer analyses of disconnected and aggregated networks. Second, we aim to assess the relative contribution of each disperser species to the cohesion of seed dispersal across habitats. We will do so by exploring dispersers multilayer versatility, which expresses their contribution to the mobile link function both within and between habitats. We discuss the potential of this new metric by comparing it to the information provided by traditional species-level descriptors.

## Results

**Overview of seed dispersal in Gorongosa.** During this one year, we collected 1399 fecal samples (1174 mammal dung piles and 236 bird droppings) produced by 98 animal species, of which 508 (29%) had at least one undamaged seed. Overall, 12,159 undamaged seeds from 94 plant species were retrieved from the feces of 29 dispersers, comprising 508 links. Focal observations produced 85 further links (14% from the total), whereas camera traps contributed with 15 new links (2.5%). In total, we compiled 608 links between 32 animal species and 101 plant species, in four habitats (Fig. 1).

Overall, primates were responsible for most interactions, namely *Papio ursinus* (chacma baboon, 35%) and *Cercopithecus pygerythrus* (vervet monkey, 10%), followed by *Loxodonta africana* (elephant, 22%) and *Civettictis civetta* (African civet, 7%) (Fig. 1). The three most commonly dispersed plant species represented 41% of all recorded interactions, namely *Ziziphus mucronata* (Rhamnaceae, 15%), *Sclerocarya birrea* (Anacardiaceae, 13%), and *Hyphaene natalensis* (Arecaceae, 13%).

We estimated that our sampling effort captured 77% of the disperser species and 44% of the plants with similar levels of sampling completeness across the four habitats (Supplementary Fig. 1 and Supplementary Table 1).

**Modular structure of the spatial multilayer network.** To evaluate the extent to which the seed–dispersal interactions are sorted into distinct communities of tightly interacting species[36], we calculated the multilayer modularity[24,37] of the spatial network of

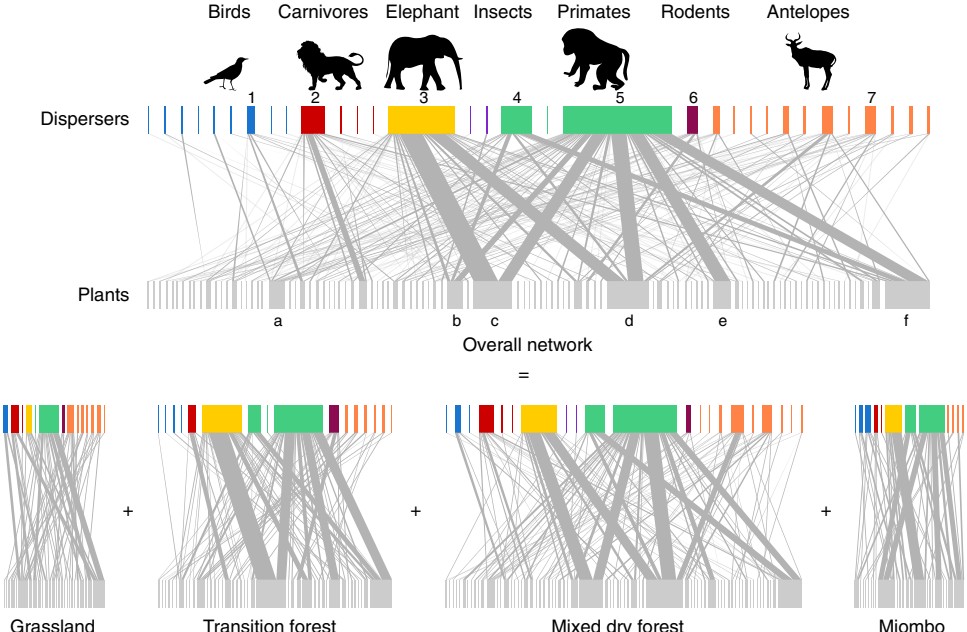

**Fig. 1** Quantitative seed–dispersal network of the Gorongosa National Park, Mozambique. Both the aggregated (top) and the individual habitat (bottom) networks are based on the same sampling effort and are represented on the same scale. The boxes in the top level represent disperser species and those on the bottom level represent the plant species dispersed. The gray lines linking the two levels represent pairwise species interactions, and their width proportional to the interaction frequency. The aggregated network was obtained by pooling all interactions across the four habitats, and summing their frequencies. Main seed dispersers: 1. *Pycnonotus tricolor*, 2. *Civettictis civetta*, 3. *Loxodonta africana*, 4. *Cercopithecus pygerythrus*, 5. *Papio ursinus*, 6. *Hystrix africaeaustralis*, and 7. *Redunca arundinum*. Most commonly dispersed plants: a *Centaurea praecox*, b *Grewia inaequilatera*, c *Hyphaene natalensis*, d *Sclerocarya birrea*, e *Tamarindus indica*, and f *Ziziphus mucronata*. The full list of species can be seen in Fig. 3 and Supplementary Fig. 2, for animals and plants, respectively. The silhouettes used in this figure are all sourced from Open Clipart and were made available under a CC0 1.0 licence

Gorongosa (see "Methods" section and Supplementary Methods for details on the multilayer modularity algorithm). Using a multilayer formalism[13], this network is defined by the animal seed–dispersal interactions (intralayer links) in each habitat (layer), with habitat connectivity (interlayer links) provided by the common species. Ultimately, interlayer links should be interpreted as the movement of matter or energy between layers, in our case the effective movement of animals and seeds between habitats, and quantified in a way that estimates the intensity of these movements (interlayer strength). Multilayer modularity was calculated across a range of interlayer strength (0–10), assuming that any co-occurring species between habitats effectively connected them with the same intensity, to test how the structure of the spatial network is affected by habitat connectivity. The multilayer modularity of the Gorongosa seed–dispersal network was very high across the whole range of values of habitat connectivity ($Q_{\text{multilayer}} = 0.903$–0.993; Fig. 2a), with an overall increasing trend toward an asymptote just below 1. We used two null models (see "Methods" section for details) to test how the structure of the spatial network is influenced by the seed–dispersal process within each habitat (intralayer null model) and by the identity of the animals connecting these habitats (interlayer null model). The structure of the empirical network, across the range of habitat connectivity values, was statistically different than that predicted by both null models, though in opposite directions: reshuffling interactions within each habitat (intralayer null model) over-estimated modularity, whereas reshuffling the identity of the habitat-connecting animals in each habitat (interlayer null model) underestimated modularity (Fig. 2a). The identity of the dispersers and the intensity of movements between habitats (inter-layer strength) play a more important role for the spatial structure of the seed–dispersal network than the pattern of seed dispersal within each individual habitat. Nonetheless, the modularity predicted by both null models tended to converge to that of the observed network at very high values of interlayer strength (Fig. 2a and Supplementary Data 1), indicating an increasing importance of random processes in structuring the networks. This suggests that when habitat connectivity is very high the overall network structure becomes less determined by the identity of animals connecting them, and might be more contingent on the structure of seed dispersal within habitats.

To understand the added value of the multilayer approach in relation to the traditional monolayer approach, we compared the results from the multilayer analysis with those provided by the currently standard approaches of either merging all data into a single aggregated network ($Q_{\text{aggregated}}$), in which interactions occurring at multiple habitats are summed across habitats, or considering each habitat as a discrete and disconnected network. The structure of the aggregated network is influenced by the distribution of the interactions among the species, with modularity being significantly lower than predicted by the intralayer null model (mean $Q_{\text{aggregated}} = 0.43$ vs. $Q_{\text{null models}} = 0.59$, $p < 0.001$; Supplementary Fig. 3). However, it ignored habitat connectivity because it cannot incorporate such information. In the disconnected network, habitats are considered totally independent from each other, thus equivalent to calculate modularity for each of them[38]. Modularity was similar or slightly higher than that of the aggregated network and much lower than that of the multilayer network, ranging from 0.43, in the Mixed forest, to 0.56, in the Grassland (Fig. 2 and Supplementary Fig. 3).

The number of modules detected in the multilayer structure is mostly constant, oscillating between 11 and 12 across most values of interlayer strength, except for very small values, where some additional modules were detected (Figs. 2b, 3, and 2). The intralayer null model consistently predicted more modules than observed, while the interlayer null model consistently predicted

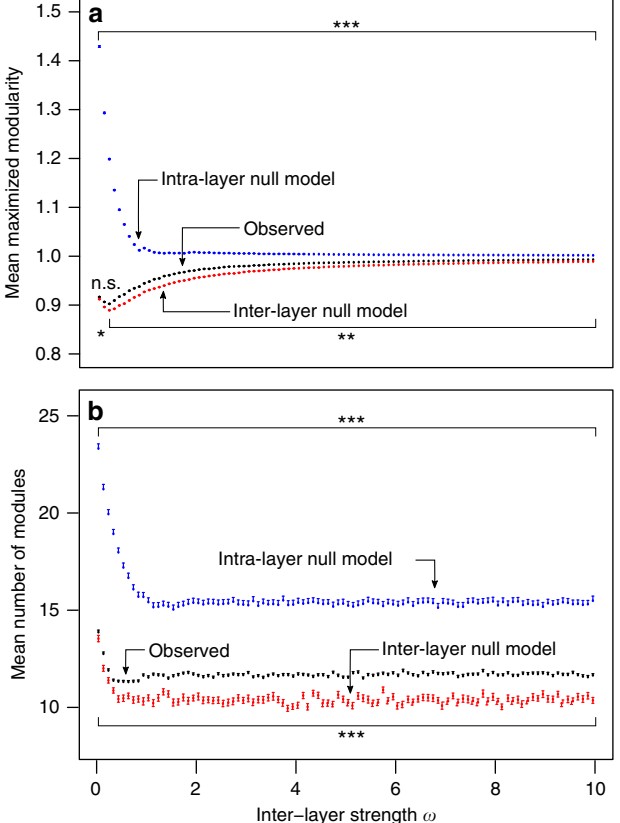

**Fig. 2** Modularity and number of modules observed in empirical networks and predicted by two different null models. Mean maximized modularity (**a**) and mean number of modules (**b**) of the observed networks (black) across the range of interlayer strength (0–10), and comparison against the two null models: intralayer null model (blue), and interlayer null model (red symbols). Values presented as the mean (±SEM) of 100 runs of the modularity function, for each interlayer strength. The significance of the observed modularity and number of modules was compared against those of the null networks with a one-sample $t$ test. $*p < 0.050$, $**p < 0.010$, $***p < 0.001$, n.s. non-significant. Full results are presented in Supplementary Data 1

fewer modules than observed, across the whole range of interlayer strength (Fig. 2b and Supplementary Data 1). In the aggregated network, the average number of modules detected was 11.4, which was in line with those detected in the multilayer network (Supplementary Fig. 3 and Fig. 2), and significantly higher than those predicted by the intralayer null model (mean modules observed = 11.4 vs. null model = 9.2; $t(99) = -28.60$, $p < 0.001$). The mean number of modules in each habitat of the disconnected network was variable and ranged from 6 to 13 (Supplementary Fig. 3 and Fig. 3). In the spatial multilayer network, modules are subsets of species that strongly interact across the different layers of the network[25,39]. For animals, this corresponds to species that occur and disperse seeds from the same plant species in more than one habitat (Fig. 3 and Supplementary Fig. 2). For example, most primates (baboon, vervet monkey, and *Otolemur crassicaudatus* (bush baby)) all disperse *Z. mucronata* and are consistently placed in the same module in the multilayer and in the aggregated networks, but not when habitats are weakly connected or considered independent. It is worth to note that module affiliations do not necessarily group phylogenetically related species, but species that feed on similar resources, which in seed dispersal might be mostly determined by behavioral and morphological constraints (e.g., *Corythaixoides concolor*, the go-

away bird, is consistently assigned to the same module of the bush babies, Fig. 3).

The strength of each interaction can vary across habitats, reflecting different animal resource preferences in different contexts, and therefore, species can change their module affiliation between habitats[24,38]. We calculated species adjustability as the proportion of animal or plant species that switch module affiliation at least once between any pair of habitats[12]. Most species do not change module affiliation across habitats, exhibiting a relatively low or non-existing adjustability (Supplementary Fig. 4). When the intensity of species movement between habitats (interlayer strength) is low, animals and plants tend to interact with distinct set of species in each habitat and a higher proportion of species will change their module affiliation between habitats. As the intensity of these movements intensifies, and habitat connectivity increases, species adjustability becomes negligible and interactions tend to occur among the same species across all habitats. However, this stabilization on interaction partners happens at different levels of habitat connectivity for animals and plants (Fig. 3; Supplementary Fig. 4; and Supplementary Data 1). For both animal and plant species, adjustability was generally more affected by the identity of the animal (interlayer null model) than by the pattern of interaction within habitats (intralayer null model). For low interlayer strength, animal adjustability was significantly lower than predicted by the intralayer null model, but higher than predicted by the intralayer null model (Supplementary Fig. 4). However, both null models performed better at greater values of interlayer strength. The interaction pattern within habitats (intralayer null model) had a variable effect on plant adjustability: the observed plant adjustability was significantly higher for very low, but also for high habitat connectivity, but lower observed adjustability between these values. The interlayer null model consistently predicted significantly higher plant adjustability for lower interlayer strengths (Supplementary Fig. 4). Thus, animals are more likely to disperse the same plant species across habitats than plant species are to rely on the same dispersers, and animal movement across the landscape exerts a stronger influence in the spatial structure of the seed–dispersal network.

**Contribution of disperser species to seed dispersal cohesion.** We did not detect differences on animal species richness across the four main habitats of Gorongosa (G test: $G_3 = 1.84$, $p = 0.61$; Fig. 4a and Supplementary Table 2). Mixed forest holds a greater richness of plants than the other three habitats, but this was only significant in comparison to Grassland and Miombo (Fig. 4b and Supplementary Table 2). As for richness of interactions, Mixed forest had more interactions than Transition forest, and both habitats had more interactions than Grassland and Miombo (all pairwise G tests: $p < 0.002$; Fig. 4c and Supplementary Table 2). Dispersers' specialization did not differ significantly among habitats ($X^2 = 2.49$, d$f = 3$, $p = 0.49$; Fig. 4d and Supplementary Table 3).

We calculated each disperser multilayer versatility, which is equivalent to an overall measure of centrality to identify those that are topologically important to the structure of the spatial network[29]. For this effect, we used a unimodal projection of the network, in which two animal species are connected if they disperse the same plant species[40], thus providing an insight over their likely "functional redundancy"[41]. Links between species were quantified by weighting the number of shared interactions by the assemblage size[42,43], minimizing the loss of information associated with unimodal projections[44]. Multilayer versatility revealed that few dispersers are disproportionately important, namely the baboon and the elephant, followed by a long tail of species with

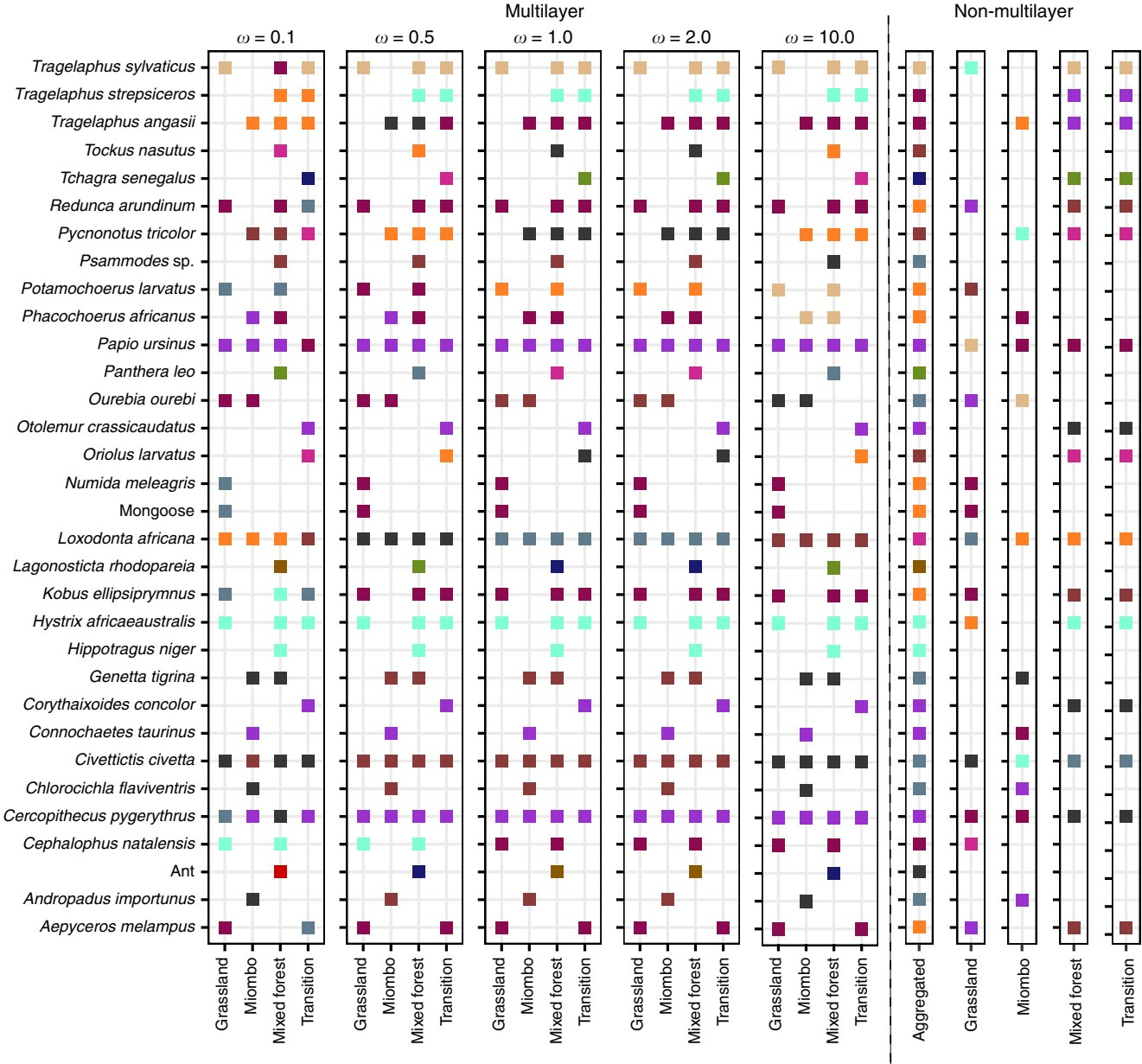

**Fig. 3** Module affiliation of animal species in the spatial multilayer network of Gorongosa. Module affiliation is shown for five different interlayer edge strengths (left block), and for the monolayer networks, considering either the aggregated network or each habitat individually (right block). In each case, the run with the highest maximized modularity was used (module affiliation for plant species is shown in Supplementary Fig. 2). Within each network, different colors represent different modules. Colors in different blocks are independent

lower versatility (Fig. 5a). The importance of these species comes from being central in the structure of the seed–dispersal network because they share plant partners with many other animals, but also because they share plant species across different habitats. The versatility of dispersers in the multilayer network was correlated with their versatility in the aggregated network ($r_s = 0.671$, $p < 0.001$; Fig. 5b and Supplementary Table 4), but the importance of species with low versatility is underestimated in the aggregated network (Fig. 5b). There were relatively few shared links among habitats (total edge overlap = 8.2%). However, all habitat pairs, except Miombo and Grassland, shared more than 20% of the interactions (Fig. 6).

We evaluated if the information condensed by multilayer versatility could be captured by other species-level metrics, namely specialization $d'$, number of habitats, and species multistrength. We did not find a significant correlation between

multilayer versatility and both dispersers mean specialization ($d'$) and the number of habitats where they occur ($r_s = -0.255$, $p = 0.208$, Fig. 5a; $r_s = 0.383$, $p = 0.053$, respectively, Supplementary Table 4). However, dispersers multistrength was only moderately correlated with their importance (i.e., versatility) on the multilayer network ($r_s = 0.514$, $p = 0.007$; Supplementary Table 4). Species multistrength extends the concept of its monolayer counterpart, expressing the total number of links of a species across all layers of the network[19], i.e., the total shared interactions with all its neighboring species across the habitats. However, contrary to versatility, multistrength does not account for the distribution of these links in relation to the other species, or the number of layers in which these links occurs. Thus, although both metrics are related, multistrength will not reflect the importance of a species for the overall structure of the multilayer network as much as versatility.

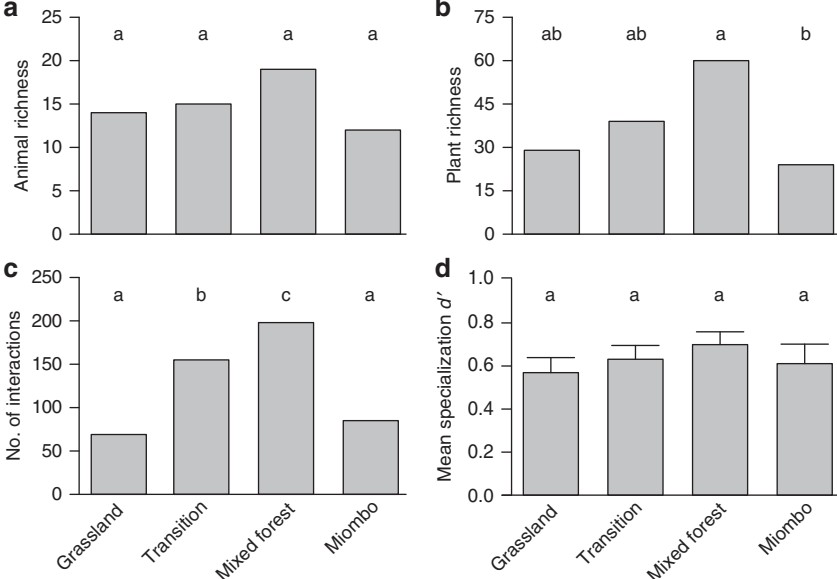

**Fig. 4** Network- and species-level descriptors of the interactions from each habitat of Gorongosa. Differences among the main habitats of Gorongosa in terms of **a** animal species richness, **b** plant species richness, **c** number of interactions, **d** species specialization (mean ± SEM). Different letters indicate statistical pairwise differences: **a–c**, pairwise G tests (see Supplementary Table 2 for full results); **d**, generalized linear mixed models (63 observations/ occurrences of 32 animal species in four habitats; Supplementary Table 3 for full results)

## Discussion

Species and communities are not randomly distributed across the planet, but they are strongly structured by spatial attributes traditionally recognized by ecologists (e.g., niches, habitats, landscapes, and biomes). Traditionally, species interaction networks have been studied as discrete entities with borders defined by the researchers based on different landscape attributes. However, species interactions do not abruptly finish at habitat borders, and therefore the decision of merging or segregating data from these spatial units is far from trivial. Nevertheless, ecologists are still faced with a paucity of tools to evaluate when such combination of data is useful, or when it might increase the noise around the patterns of interest, thus obscuring important conclusions. Although still based on the recognition of different habitats, the implementation of a multilayer approach provides a valuable tool that allows for better decisions regarding the merits of segregating or merging spatially (or temporal) explicit data.

However, interlayer connectivity has never been explicitly incorporated in the analysis of the modular structure of spatial ecological networks. In this study, we investigate the spatial structure of a seed–dispersal network spanning across multiple habitats explicitly considering interlayer connectivity. We made use of a highly comprehensive data set collected in a highly diverse African landscape including all potential disperser guilds. This adds to the sparse knowledge of seed dispersal in Africa, but has direct implications for our understanding of seed–dispersal networks across the globe.

Our results show a highly modular structure of the spatial multilayer network that is influenced by the strength of the connectivity between habitats, with about half the communities of seed dispersers detected bridging most of the habitats (Figs. 3 and 4). The network is dominated by a few highly versatile species that secure both local (habitat level) and global (landscape level) dispersal of seeds, ensuring the spatial continuity of the seed–dispersal process.

Landscapes are intrinsically dynamic, being constantly shaped by local disturbance and ecological succession[45]. Understanding how animals move between habitats providing key mobile links[10], and how ecological interactions are distributed across

habitats[15,16,18], has long been recognized as critical for the dynamic of patchy habitats across complex landscapes[35]. Nevertheless, and despite the current interest on species interactions networks, these are yet to explicitly accommodate this interlayer dynamic when analyzing the structure of spatial networks.

Here, we implement for the first time a multilayer approach to evaluate the spatial structure of an ecological network explicitly incorporating the interlayer strength connecting networks from adjacent habitats. Our spatial multilayer seed–dispersal network exhibited a highly modular structure, i.e., species tend to interact with subsets of species (i.e., modules) within subsets of spatially coupled habitats. By explicitly including non-zero interlayer links, i.e., the habitat connectivity promoted by the common species, it is possible to account for the interdependence of the network structure across multiple habitats[38], and identify modules that spread across habitat borders[25].

For a more realistic module detection, interlayer strength should ideally be measured empirically to reflect the effective movement of the individual species across habitat borders[15]. Unfortunately, obtaining such data at the community and landscape levels, i.e., all species, across all habitats can be incredibly challenging. The alternative of assigning the same interlayer strength to all species, i.e., assuming that all species connect habitats with the same intensity, is a clearly undesirable simplification as species connect habitats with different intensities because of their differential ability to move across and establish in a given habitat[46,47]. Incorporating such empirical data could have important implications on modules found by the modularity function; the relative importance between intra- and interlayer process would be different for each individual species, thus affecting its probability of changing module affiliation[38]. Exploring the modular structure across a range of interlayer edge strengths is an alternative to obtaining empirical data, and has been often done in other fields to understand its importance for processes spanning across different layers[24,25,38,48]. Our analysis revealed that the modularity and the number of modules are mostly affected at extremely low levels of interlayer strength, suggesting that the spatial community structure can be maintained even if the strength of the habitat connectivity is low

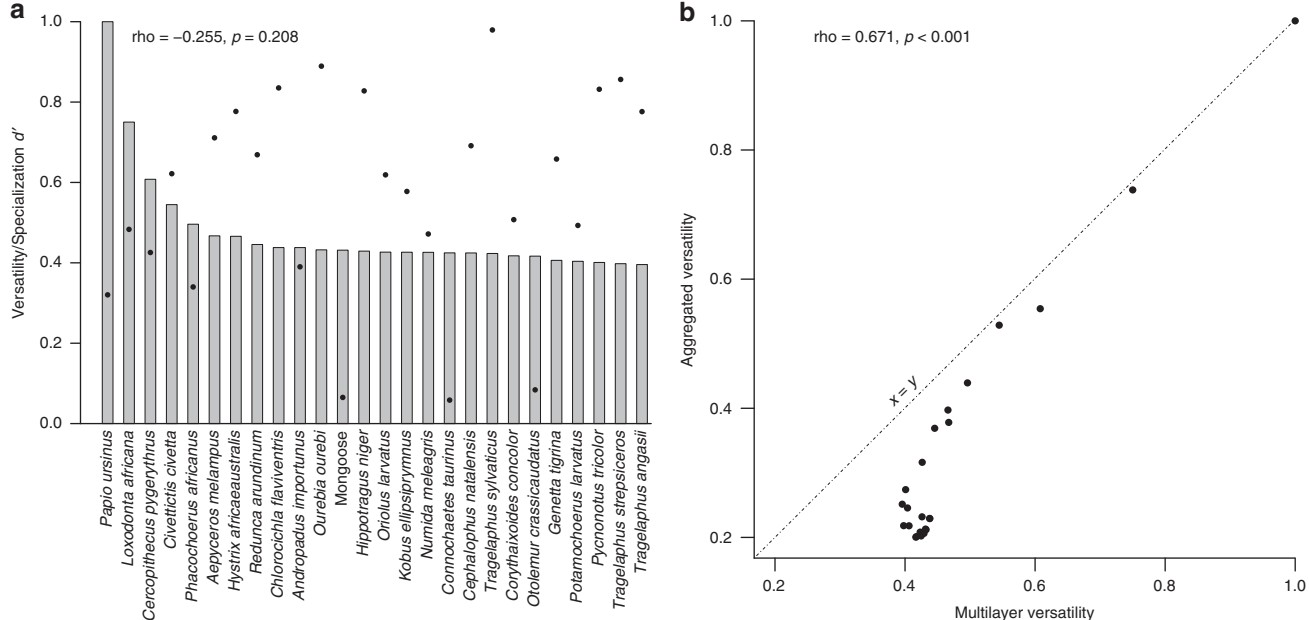

**Fig. 5** Correlates of multilayer versatility. Correlation between animal species versatility in the multilayer network (bars) and the mean specialization d′ (dots) (**a**), and between multilayer versatility of the monolayer versatility of the aggregated network (**b**). Full data available in Supplementary Table 4

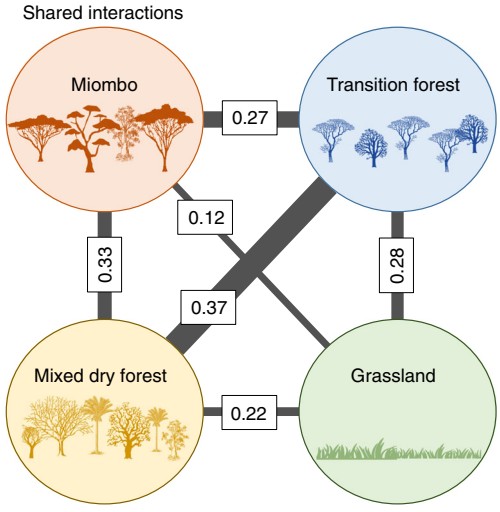

**Fig. 6** Similarity in terms of shared interactions, between the habitat pairs of Gorongosa. Similarity is assessed using the edge overlap between the habitats in the spatial multilayer network, i.e., proportion of shared links across habitats. The width of the links is proportional to the correlation between each habitat pair. The silhouettes used in this figure are all sourced from Open Clipart and were made available under a CC0 1.0 licence

relative to the strength of the interactions within the habitats. The inter- to intralayer edge strength ratio is essential for the outcome of modularity estimation, affecting the probability of nodes being assigned to distinct modules[38,48]. As such, when interlayer coupling increases, the number of detected modules is expected to decrease[24,25]. Importantly, the structure of the seed–dispersal network was not fully captured by the aggregated network or by considering each habitat as an independent network. Consequently, the result obtained by using a multilayer approach is not biased by any decision regarding aggregating or disconnecting the different layers of the network. Instead, the resulting structure is a consequence of the relative importance of the processes occurring

within and between layers, which is objectively defined by the relative strength of the inter- to intralayer edges.

Regarding the modules' composition, these grouped-together species that are not always phylogenetically close (e.g., primates were grouped with the go-away bird), suggesting that functional and morphological matching, such as gape-size and seed/fruit size, are more important drivers of seed–dispersal interactions[40]. Interestingly, we detected low adjustability for most species and module affiliations remained mostly constant across habitats. Module switching occurred only for some species (e.g., primates, elephants, or civets), and at very low values of habitat connectivity (Fig. 3). Most animals, however, tend to disperse the same plant species in different habitats, thus maintaining a similar functional role across the landscape[26], even if habitat connectivity is very low. This can only be detected if the habitats are explicitly linked in the analysis of network structure. Resource availability largely determines animal movements at the landscape level[17]. In turn, the inter-habitat movement of dispersers is likely to affect plant regeneration dynamics, and thus resource availability. The capacity of species to adjust their interactions to specific contexts (thus increasing overall adjustability) is likely important for species persistence in changing environments, while at the same time tends to promote a greater connectivity (e.g., seed dispersal) across habitats. In Gorongosa, some of the species that changed module affiliation have generally wide-range movements and can distribute seeds between habitats, thus giving a key contribution to plant genetic diversity and spatial distribution of plant populations through seed dispersal[34,49]. This is particularly true for intrinsically large-scale processes, such as seed dispersal, while other processes, such as belowground mycorrhizal associations, seem to be structured at much finer scales[50].

Although the number of interactions differed among habitats, and the overall edge overlap was relatively low, neither the richness of dispersers, the mean number of dispersed plants, or dispersers' specialization varied significantly.

Multilayer versatility allowed us to identify the animal species that are most important for dispersing seeds simultaneously at the

local/habitat scale and at the global/landscape scale by linking multiple habitats. In the Gorongosa landscape, primates, elephants, and African civets are central nodes in the network and likely important for its stability and cohesion[51]. Although the versatility of the aggregated network can correctly identify the most important dispersers, it underestimates the importance of species that are restricted to one or a few habitats[29] (e.g., *Genetta tigrine* (genet) or *Tragelaphus strepsiceros* (kudu); Supplementary Fig. 5). The relatively low correlation between multilayer versatility and multistrength, and the non-significant relationship with the number of habitats where each species occurs and its specialization (*d′*) reflect the information gain of using multilayer versatility, which could not be captured by conventional metrics.

The most versatile seed dispersers of Gorongosa were those switching module affiliation between habitats. They are known for incorporating a high proportion of fruits into their diets[52–54], having relatively long gut retention times, and traveling for long distances[53–55]. This allows seeds to escape the high intraspecific competition near their parent plants by diversifying the deposition site of ingested seeds, and thus increasing their chances of successful recruitment[56]. Baboons are ubiquitous in Gorongosa[32] and assume a central role as seed dispersers across the whole park, and elephants, whose populations are still recovering in Gorongosa[32], are also essential to the dispersal of plant species, particularly those with very large fruits and seeds (e.g., *H. natalensis* or *Borassus aethiopum*)[57]. Surprisingly, despite being locally abundant and often considered important seed dispersers in many ecosystems[58], birds had a low versatility. While bird versatility might have been underestimated as a result of the different sampling method used (mist-netting), the low proportion of bird droppings with seeds (only 7 out of the 96 bird species captured where found to disperse seeds, and these were only found in 19 out of the 236 bird droppings analyzed), and the consistently low sampling completeness estimated to all sampling methods (transects: 25%, mist-netting: 13%, and focal observations: 13%) suggests that the lower bird versatility is actually structural rather than a sampling artifact. Potential biases may emerge if any particular animal or plant groups are under or oversampled due to the use of different sampling methods. This problem might be countervailed by performing analysis on rarefied or unweighted networks, though these are subjected to their own caveats. We are only aware of a single study that explored the potential consequences of merging data originated from different sampling methods in the assembly of seed–dispersal matrices[59]; the authors concluded that this was in fact beneficial due to the complementarity of the methodologies. Moreover, it must be noted that the consistently low estimates of sampling completeness are likely to be largely underestimated for at least two reasons: first, because species accumulation curves are based on the assumption that communities are closed, an assumption that is not met in year-round studies, where new species "enter" the community of potential interactions as a consequence of advancing phenology (i.e., fruiting season); and second, because a large proportion of the potential interactions will never be detected because they are not really possible due to spatial, temporal, and phenological mismatch between species[60,61]. These "forbidden links" (or "true zeros") can amount to a large proportion of the unobserved potential links (up from 44 to 77%) in seed–dispersal networks[61].

Interestingly, the significant edge overlap between habitat pairs (i.e., the proportion of shared links) confirms the a priori assumption that seed dispersal does not stop at habitat borders. Taken together with the results from multilayer modularity, showing that species tend to maintain their module affiliation throughout the landscape, these results suggest that species traits (such as mobility and size) may largely determine their multilayer versatility.

The effective conservation and restoration of natural areas requires an integrated view of how species and their interactions maintain functional ecosystems on complex landscapes, and a multilayer approach is a most valuable tool to explore these factors. Here, we took a step further in the analysis of spatial mutualistic networks, and using interlayer edges strength, we explicitly considered the interactions between plants and their dispersers across multiple habitats in the analysis of the network structure. Furthermore, we identified key spatial coupler species, which play a pivotal role in long-term vegetation dynamics in Gorongosa by ensuring the dispersal of genes across the landscape. These key spatial couplers, namely primates, elephants and African civets, should be highly regarded in the protection of this essential ecosystem service.

As many other types of networks, mutualistic networks are not temporally static, and they do not abruptly stop at habitat borders[7]. Therefore, forcing the analysis of biotic interactions into spatially delimited "network snapshots" and ignoring habitat connectivity will inevitably limit the insights that can be gained by this approach[8]. Here, we show that a multilayer approach can be used to link ecological processes that occur in different spatial layers of a network providing insights that may not be captured using traditional representations of monolayer networks. Overlooking the multiple relationships between nodes on different layers may lead to an inaccurate network structure, but also to misidentification of the real role of species in the whole-network structure[19,24,29]. Alternatively, the explicit incorporation of the temporal and spatial dynamics into a multilayer network approach represents an important next step in the study of animal-plant interactions.

## Methods

**Field site and sampling.** This work was carried out in the Gorongosa National Park, Mozambique (hereafter Gorongosa), a hyperdiverse park[62] covering 4067 km² at the southern end of the Great Rift Valley (18°47′43.2d″S 34°28′09.1″E). We defined four major habitats based on the vegetation structure and flooding regime: (1) grassland, periodically inundated grassland, with few shrubs and virtually no trees; (2) Transition forest, characterized by short flooding periods and dominated by trees of *Faidherbia albida*, *H. natalensis*, or *Acacia xanthophloea* with a mostly open understory; (3) Mixed forest, occasionally flooded and formed by a diverse mixture of tree species with a dense and closed understory; and 4) Miombo, one of the most extensive habitats in Africa, unaffected by floods, and dominated by *Brachystegia* spp. trees with a dense understory (see ref. 50 for details). Throughout a year, we reconstructed seed–dispersal interactions from the four habitats by retrieving intact seeds from animal dung collected in the field. Sampling took place in 12 occasions, evenly spaced between June 2014 and May 2015, except from December to February when floods make the park mostly inaccessible. In order to sample all potential disperser guilds, we employed complementary sampling protocols. In each sampling occasion, one transect ca. 2000 m and 5 m wide was performed in each habitat and separated from any other transects by at least 350 m. Overall, 48 transects were run (ca. 96 km), corresponding to a surveyed area of 480,000 m². A team of two observers simultaneously collected animal dung samples, corresponding to the deposition of a dung pile of a single animal, and identified the disperser species by direct observation of defecating animals, or using the expertise of local park rangers and field guides[63,64], and recorded direct observations of animals ingesting fruits. Bird dispersal was evaluated by collecting droppings from birds captured during mist-netting sessions on each sampling occasion, run for 5 h after dawn. Birds were kept inside individual holding bags, and released after producing a dropping[65]. All samples were carefully screened under a ×40 magnifying microscope, and all undamaged seeds identified against a reference collection of seeds/fruits collected in the field. Seeds that could not be identified visually were barcoded, and their DNA sequences compared against online databases, with species identified based on the best match and on a checklist of Gorongosa plants[33] (see Supplementary Information for details). Most seeds (92%) were identified to the species level, 7% to genus level, and less than 1% to the family level or grouped into morphotypes, hereafter referred to as "species" for simplicity. Sampling was further complemented with the analysis of motion-triggered camera traps, operating for five nights per habitat per sampling occasion to record interactions from non-conspicuous animals or those feeding at night. We estimated sampling completeness for animal and plant species in each habitat. The use of different methods to record interactions could result in different interaction sampling completeness, thus we also estimated completeness of interaction sampling for each method. This was done by estimating the proportion of observed

richness in relation to the total asymptotic richness estimated by the non-parametric estimator Chao2[66] using function *specpool* from package vegan[67], in R software[68].

**Multilayer network construction**. We assembled a multilayer quantitative bipartite seed–dispersal network for each habitat, which were visualized using the R package bipartite[69]. Interaction frequency was quantified by calculating a pooled frequency of occurrence, collating the information from the different sources (fecal analysis, mist-netting, camera traps, and direct observations). This pooled frequency of occurrence, resulted from the direct sum of all fecal samples containing at least one seed of a given plant species[5,33,70], the number of transects where a given focal interaction was detected, and the number of camera-trap recordings (1 night = 1 sample), where a given interaction was detected.

Using a multilayer formalism[13], we assembled a multilayer network formed by: (a) a set of nodes (called "physical nodes") representing the animal dispersers and the plant species whose seeds are dispersed, (b) a set of layers representing the different habitats; (c) a set of "state nodes" that correspond to the manifestation of each node on a given layer; and (d) a set of two types of edges connecting the nodes pairwise, namely, intralayer edges (i.e., animal seed–dispersal interactions); and interlayer edges, connecting state nodes between layers (i.e., animal or plant species between habitats). Interlayer edges encode the movement of animal and plant species between habitats. While the temporal scale of the movements is not exactly the same, both animal and plant genes frequently cross and establish in neighboring habitats[71], and for that reason can be considered effective habitat connectors with the strength of the interlayer links quantifying the intensity of the habitat coupling provided by this movement. Spatial multilayer networks have a categorical (non-ordinal) coupling, i.e., interlayer edges are not constrained in any specific order and any pair of layers can be connected[13]. The quantification of the interconnectivity of the multilayer network is used in the calculation of some (modularity and multistrength), but not all the network diagnostic (versatility and edge overlap).

To assess differences in the richness of animals, plants, and plant–animal interactions among habitats, we used a *G* test. If an overall effect was present, we performed pairwise *G* tests to identify differences between habitat pairs. This analysis was performed with the R package *RVAideMemoire*[72].

**Modular structure of the spatial multilayer network**. To evaluate the extent to which the seed–dispersal interactions of Gorongosa are sorted into distinct communities, we calculated multilayer modularity ($Q_{multilayer}$). Multilayer modularity, as its application to monolayer counterparts, quantifies to what extent nodes are organized into modules of strongly interacting nodes interacting more frequently than expected by chance[21]. The *Q* modularity function was maximized applying a "generalized Louvain" method[37]. This method proceeds until the network configuration that maximizes the weight of the edges inside the modules in relation to a null model is found (see Supplementary Information for details).

Following Pilosof et al.[12], the modularity function was adapted to reflect the bipartite nature of our networks. The "generalized Louvain" method requires the specification of two parameters: the resolution limit γ, and the interlayer coupling ω. The resolution limit defines the detail to which the network will be resolved into communities, and can be viewed as the importance given to the null model compared to the empirical data[38], and we used the default resolution γ = 1[24,38]. The interlayer coupling quantifies the strength of the connection between layers of a network, i.e., the effect that species have in connecting the different layers. However, measuring such interlayer strength is intrinsically challenging[12] and there is no absolute method to do so. To explore the importance of interlayer strength for the detection of modules, we calculated modularity along a range of values of interlayer strength (from 0 to 10), assuming uniform interlayer strength across all species, i.e., all species connecting any pair of habitats have the same effect in the interlayer process[38]. The stochastic nature of this algorithm means that a different maximum is reached on each run, thus we run the modularity function 100 times, and averaged the results obtained[12,25]. We compared the results obtained with a multilayer network to that of two different representations of the same network: (a) an aggregated network ($Q_{aggregated}$), where all interactions across the different layers were pooled to create one overall aggregated network, with the frequency of interactions that occur in multiple habitats being summed, and (b) a disconnected network where habitats are considered fully independent from each other, i.e., interlayer strength is set to zero, and thus modularity is calculated for each of them[38].

We test the modular structure of the observed network comparing it to the structure of networks built under the assumptions of two null models[12,25]: (1) an "intralayer null model" that keeps the number of links (i.e., connectance), while redistributing the individual interactions across the whole matrix as implemented by *r00_both* in vegan[67], was used to assess the influence of structure within each layer/habitat in the modular structure of the multilayer network[25]; and (2) an "interlayer model" following the same rationale of the *nodal* model[25], i.e., keeping the same matrix, but redistributing species identities independently in each layer, to assess if the modular structure is dependent on the identity of the species connecting the habitats. We run each null model 100 times along the same range of interlayer strength (0–10). The significance of the observed modularity was

compared against the distribution of the modularity of the null networks, and presented as the proportion of networks generated by the null models with modularity lower than the observed networks. For each network, we calculated the mean number of modules and the mean adjustability[12] of animal and plant species as the proportion of species in each level of the network that changed module affiliation at least once between habitats. The mean number of modules and adjustability of the observed networks was compared against that of the networks generated by the null models with a one-sample *t* test[12,25].

Multilayer modularity calculations and the interlayer null model were performed in MATLAB[73].

**Contribution of disperser species to seed dispersal cohesion**. First, we assessed whether the specialization of seed dispersers differed consistently between habitats by calculating animal specialization (*d′*), which quantifies their selectiveness for seeds within the range of resources used[74]. However, the number of interactions of a species is considered to reflect both resources availability and consumer activity. This metric takes into account the pattern of interaction of a species in relation to the available resources, while being robust to sampling effort, network size and asymmetry[74], also see ref. [75]. We used a GLMM, with Gamma errors, and included disperser species as a random factor. The model was analyzed with the R package *lme4*[76]. If any level of the independent variable was significant, we assessed its overall effect with a Wald $X^2$ test available in R package *car*[77]. The overall fit of the model was assessed with the Akaike's information criterion against a reduced model, which only included the intercept. Pairwise comparisons between habitats were tested with Tukey HSD test with the R package *multicomp*[78].

Secondly, we explored seed–dispersers' importance in the network by calculating their overall versatility as seed dispersers. Versatility identifies species that are topologically important for the structure of the multilayer network[29]. Versatility was calculated using software muxViz 1.0[79], which adapts the Google's PageRank algorithm[80] to describe the position of a node within the structure of a network based on a random walk between adjacent nodes[29] (see Supplementary Information for further details), being equivalent to a global measure of centrality. The implementation of this method requires bipartite networks to be projected onto unimodal networks. While some projection methods entail some loss of information[44], we applied a weighted projection which estimates interaction weight based on the proportion of shared interactions (i.e., seed species shared by disperser species) relatively to the total network size[42,43], thus minimizing the loss of information. The projection was performed with function *projecting_tm* from the R package tnet[44]. This algorithm is particularly suitable to multilayer networks[29] as it condensates information on dispersers niche overlap[40], based on the importance of their shared dispersed seeds[29].

To understand how animals shared their links across the different habitats, we calculated edge overlap with software muxViz 1.0[79], which quantifies the proportion of common links between animals across habitats.

We evaluated the information condensed by multilayer versatility in respect to the species versatility of an aggregated network, and to other species-level metrics, calculating its correlation with aggregated species versatility, specialization *d′*, multistrength, and number of habitats in which a species is present. The specialization index d′ measures the distribution of a species interactions with each partner over the total number of partners available[74]. Multistrength is an extension of its monolayer version, and sums the total weight of the links incident on a node across all layers, taking into account the links connecting nodes in the different layers[19] (see Supplementary Information for details). It expresses the total number of shared interactions of a dispersers species with all its neighboring species across all habitats. Versatility and multistrength were calculated using muxViz 1.0[79].

**Code availability**. MATLAB scripts for the estimation of modularity and generating the interlayer null model are available from https://doi.org/10.6084/m9.figshare.4955651, and R code for generating the intralayer null model, analysis of the modular structure, and generating the files to be used with muxViz are available from https://doi.org/10.6084/m9.figshare.4836383.

**Data availability**. The seed–dispersal interaction network matrices are available on reasonable request.

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

## Acknowledgements

This work was supported by FEDER funds from the "Programa Operacional Factores de Competitividade – COMPETE" and by national funds from the Portuguese Foundation for Science and Technology – FCT through the research project PTDC/BIA-BIC/4019/ 2012. S.R.-E., R.H., and M.C. were also supported by FCT (Grants IF/00462/2013, IF/ 00441/2013, and SFRH/BD/96050/2013, respectively). We thank Greg Carr, and the Greg Carr Foundation – Gorongosa Restoration Project, Dr Marc Stalmans and all the staff from Gorongosa National Park for their logistic assistance during fieldwork, and for sharing their knowledge and passion about Gorongosa.

## Author contributions

S.T., R.H., H.F., and S.R.-E. designed the study; S.T. and M.C. collected the data; S.T. led the analysis and wrote the first draft; and all authors contributed to manuscript revisions.

## Additional information

**Competing interests:** The authors declare no competing financial interests.

