## [Peer Review File · Nature Communications]

Reviewers' comments:

Reviewer #1 (Remarks to the Author):

This interesting study applies multilayer network analysis to an impressive seed-dispersal dataset collected over the span of an entire year in Gongorosa National Park, Mozambique, that supports diverse populations of animal dispersers, including insects, birds, and mammals. Two aspects of the study really stand out. The first is the dataset, which is unusual in terms of the breadth of interactions it encompasses (there was an effort to identify the interactions of the “entire” seed-dispersal community, rather than arbitrarily circumscribing the types of species investigated) and its geographic context (seed dispersal remains poorly studied in African savanna and woodland habitats relative to tropical forests, and especially Neotropical forests). A second noteworthy aspect of the study is the use of multilayer network analysis, which has not yet been widely applied in ecological contexts (see also Pilosof et al. 2015, ref. 13 in the manuscript). This approach enables researchers, at least in theory, to analyze the spatial and temporal structure and dynamics of complex networks (for example, to analyze the linkages between different habitats in a landscape, as was done here).

The authors identify 13 distinct “seed-dispersal [sub]communities” in a highly modular overall network structure, many of which occur in more than one of the four habitat types analyzed (grassland, transition forest, dry forest, miombo; their Table 1 and Fig. 1). This crossover of communities across habitat types leads to a large number of interactions and network hub species that are shared across habitat types (Fig. 2); although the diversity of plant species, animal species, and plant-animal interactions varied across these four habitat types, the mean normalised degree and specialization of dispersers did not (Fig. 3). However, disperser specialization and versatility varied considerably among disperser species (Fig. 4).

I very much appreciate the work done here and think that there is the potential here for an important contribution to the literature. However, I think some important work remains to be done to achieve this. The novelty of the study in terms of data and approach is fairly easy to identify, but I am struggling a bit to apprehend the novelty, robustness, significance of the biological insight. When boiled down, the biological conclusions summarized in the previous paragraph seem to be either (a) not particularly surprising and/or (b) interpreted and presented at a fairly superficial, phenomenological level without much attention to biological mechanism. Put another way, although I appreciate the value of looking at biological phenomena in new ways, it is not clear how our understanding of this problem has really been substantively advanced by this new way of looking at it. I would challenge the authors to explain how their results challenge conventional understanding of seed dispersal, if indeed they do.

For example, consider the 13 seed-dispersal communities in Table 1—a result that seems to me to be entirely phenomenological, with no robust (or even hypothesized) insight into the biological mechanisms that would generate these groupings. Community #6 consists of two viverrid carnivores, a small antelope, a tenebrionid beetle, and two pycnonotid birds, whereas community #8 comprises only porcupine and sable antelope. What do these groupings mean, and how/why do they arise? Or are they rather an artifact of the methodological approach? I myself am at a loss to explain these groupings in light of the respective ecologies, diets, and behaviours of the species involved; it seems to me far more likely that they are curious artifacts of sampling and analytical approach than that they reflect something biologically “real” that totally transcends the apparent natural history of the system as understood by me and many others. If the authors can convincingly explain these results, then that would be very interesting indeed, but it’s not clear that the data are up to this task.

Similarly, the finding that the four habitat types share many interactions and hub species (Fig. 2)

is not particularly surprising given that many species occur in multiple habitat types (Table 1, Fig. 1) and that the most important dispersers (e.g., baboons) are described as “ubiquitous.” Even here the authors’ mechanistic interpretation of the results is not entirely clear. One of the authors’ reasons for using multilayer network analysis is that it avoids researchers’ having to make arbitrary decisions about who/what to include in the network, but the delineation of habitat boundaries itself seems like an example of this kind of semi-arbitrary distinction. So, does the generally high “multilayer versatility” of dispersers across habitat types truly reveal something real about the “spatial cohesion of seed dispersal,” or does it simply reflect that the researchers have defined the “habitat borders” in an artificial way (or in a way that is incongruent with the actual biology of the system)? Might it be more unambiguously informative to consider space as a continuous quantitative variable, as opposed to these loosely defined categories?

And finally, what are we to make of the among-species variation in versatility and specialization (Fig. 4)? Is this a function of differences in the species’ diets? Of differences in their foraging behaviour, mobility, and home range? Of differences in their relative abundance? I suspect the answer is “all of the above,” but without knowing anything about the relative contribution of these different factors, it is difficult to derive any expectation about the extent to which these results will be generalizable. Can the authors’ specify any testable predictions arising from their results that future studies could address to reveal whether the network properties revealed here are fundamental vs. idiosyncratic?

Data quality is another factor contributing to my uncertainty about the validity and significance of these results. As noted above, I think this is a very impressive dataset in many respects, and I commend and congratulate the authors for their ambitious attempt to characterize the entire seed-dispersal community (although remember that small mammals can be very important seed dispersers), for doing so across an entire year in multiple sites, and for confirming the identity of difficult-to-identify seed types using DNA barcodes. However, there are many factors that could still profoundly influence the results of this kind of analysis, which I do not think have been adequately acknowledged or accounted for. These include: (i) potential misidentification of dung types, because ranger IDs and field guides are unreliable for many species with similar-looking dung—perhaps dung be identified using DNA barcodes, similar to what was done with plants?; (ii) bias towards large-seeded plant species, as small-seeded species are notoriously difficult to detect in the absence of complementary germination trials; (iii) sampling biases leading to undersampling of certain species (notably birds), and the related (iv) failure to account or correct for disperser species’ relative abundances and movements. With respect to this last point, on lines 393-400, the authors argue that baboons, elephants, and civets are the “most important” species for “enhancing network connectivity at the habitat and landscape level,” and that “this importance would have been missed” if not for the multilayer network approach. But whereas I have no doubt that baboons, elephants, and civets are important seed dispersers in this system, it is unclear both (a) to what extent they actually *couple* habitats, since the authors do not appear to have assessed the actual movement of seeds across habitat boundaries (although I believe that this is highly likely), and (b) to what extent their inferred importance, specialization, and versatility is simply a function of their being highly abundant relative to other species and therefore having the largest number of scat samples examined. Conversely, it seems statistically inevitable that species with a small number of samples will be inferred to have relatively low importance and versatility, and relatively high specialization, simply because a small number of samples cannot reveal a large number of interactions. The analysis of completeness (Table S1) for the community at large does not seem to account for this relative variation in sampling across different species (better would be if each species’ interactions were assessed using rarefaction, but this would not be possible for poorly species represented by only one or a few samples). It seems to me that the conclusions derived from this kind of analysis are likely to be highly sensitive to variation in all of these factors, again contributing to uncertainty about the biological interpretation of the statistical patterns being reported.

In summary, I respect the work done here and I see great potential for this dataset and this kind

of analysis to generate new and influential insights into the structure of mutualistic networks. But I also think that the authors are obliged to do more to ground their analysis in the biology of the system, to acknowledge (and where possible quantitatively analyze) the potentially biasing sources of variation, and to specify what can be done to further test or refute the conclusions of this study in other systems. If we do not insist upon these criteria, then it will be impossible to gauge the extent to which fresh analytical approaches such as multilayer network analysis genuinely reveal previously unappreciated dimensions of these systems, or whether they instead generate spurious and artifactual patterns that muddle the picture and lead future investigators down blind alleys.

Reviewer #2 (Remarks to the Author):

In this paper the authors explore the structure of a bipartite animal-seed network across 4 different habitat types in the Gorongosa national park. They aim to identify cohesive groups (modules) of animals and seeds across habitats, as well as to characterize the role of animals as couplers of seed dispersal across habitats. The novelty of the manuscript lies in its multilayer network approach, which is used to quantify the structural patterns across the 4 habitats simultaneously, thereby enabling explicit consideration of the effect of habitat connectivity on the structure of the network. Overall, I find the approach novel and adequate for the question in hand. As discussed in the manuscript, this approach also has an applied value for conservation. The ecological application for multilayer networks presented here can also be of interest for non-ecologists because until now the field of multilayer networks has seen very little ecological applications. I also appreciate the tremendous effort put into data collection, which adds to the strength of this paper.

That said, the paper has some weak parts that need considerable improvement. In particular, the calculations of network diagnostics and/or their interpretation in ecological terms is incorrect or missing in some instances. I also think that the paper is not focused enough and this obscures its novelty. The authors calculate many network diagnostics, and these are not properly discussed/interpreted, while the interpretation of modularity, which is key to this manuscript is not good enough. I think it is better to calculate few diagnostics that serve to make the point and discuss them properly (see specific comments below). In particular, I would lose some of the less informative diagnostics and instead calculate modularity and versatility in an aggregated network and in each layer separately to show that the module composition and versatility change as compared to a multilayer network. This will give much more insight into the ecology of the system: What knowledge do we gain by breaking the aggregated network into habitats? What is the role of species within each layer and in the whole system? This will also strengthen the validity and novelty of the approach.

My overall opinion is that the paper has a novel and valuable idea and approach but needs to improve on the methodology and ecological interpretation, and it also needs a better focus in the application of the network approach. With such improvements the paper will be adequate for the journal. Below I detail my comments and ideas for improvements.

Good luck!

Major comments

INTRODUCTION

1. The terms "community" and "community structure" have different meanings in ecology and network science. I would define these in the beginning and use community for the ecological context, and "modularity" for the network context. (e.g., lines 97 and 205 can be especially confusing).

2. Lines 60-70: While it is correct that most ecological studies use monolayer networks, there are many many studies that use multiple networks, including some that investigate spatial structure (e.g., Trøjelsgaard et al. 2015). While the use of multiple networks does not explicitly consider interlayer connectivity (which is the novelty here), it does acknowledge the fact that network structure can change across space. This point is worth to bear in mind and also should be mentioned in the introduction. In addition, I would make the strong statements more subtle, to not offend researchers who did use multiple networks. Also, I guess that by "merging" (line 62) the authors mean aggregate? A short discussion on what is gained by considering interlayer edges, as opposed to using multiple networks (e.g., by applying Poisot's framework from (Poisot et al. 2012) or aggregating networks will make the case of the study stronger. For example, in line 70: the distribution of interactions across habitats has been studied using multiple networks and for this interlayer edges are unnecessary. Finally, the statement on artificial borders or comfort zone is, to begin with, too strong and should be expressed more subtly; and more importantly, an inherent problem of how we subjectively perceive ecological systems. There is no absolute truth here. In this study the authors also defined the borders between the habitats artificially while in practice there may be intermediate transition zones or other ways to define the habitats.

3. Lines 80-95. A short description of the ecological interpretation of these network diagnostics is in place. What is species role? What does multidegree (or its monolayer version degree) mean in ecology?

4. Objectives: what is 'spatial fidelity'? How does a modular structure serve as a proxy for spatial fidelity? For me, spatial fidelity of a species is intuitively simply the proportion of habitats in which it occurs out of possible ones. I do understand how multilayer modularity can inform on groups of species that span across landscape types. 2nd objective: what is a "mobile link function"? And I would change "landscape cohesion" to dispersal of seeds across different landscapes (I guess this is what the authors mean). I would also just use "habitats", as used later on than landscapes.

METHODS

1. Experimental design: I suggest to use species accumulation curves to determine collection completeness. I could not find any mentioning of the plant species. These should be mentioned in a dedicated table along with information on their abundance.

2. Data and code: I was missing the R and Matlab codes and data to replicate the findings. This includes the files fed into MuxViz. If someone wants to replicate the results or take the approach and implement it on other data this is crucial!

3. Please provide the equation for how you calculated interaction frequency. It is not clear enough. Especially so because it is a composite of several collection methods.

4. A clearer description of the multilayer network is needed. What does it mean that layers were

coupled by common species? The authors should state explicitly what are the interlayer edges. Do interlayer edges connect each species to itself across layers ('diagonal coupling'; see (Kivelä et al. 2014))? Do they connect animals and seeds from different layers? Also, what do these interlayer edges mean? For example, are they a proxy for animal movement? This point is crucial because the whole novelty of this paper is that it considers interlayer connectivity. Readers should have a feeling of what this interconnectivity means. It is also important to mention that this interconnectivity is used to calculate some (e.g., modularity) but not all (e.g., versatility, edge overlap) network diagnostics. Also, it is important to point out that physical entities (i.e., animal and plant species) can appear in different layers, and that different replicates of the same species in different layers are called 'state nodes'.

5. Interlayer edge weights (ω): The interpretation of the interlayer coupling is not correct. It is true that the "balance" between interlayer and intralayer edge values can greatly affect the results. However, it is incorrect that $\omega=1$ means equivalent intra- and inter-layer coupling (line 220). If this were the case, then each intralayer edge should have also existed as an interlayer edge with the same value, which is clearly not the case. Instead, the authors should state that because the values of the interlayer edges were not empirically measured, they assigned them a value ($\omega=1$), and that this value is uniform across all the interlayer edges. This assumes that each animal and seed has the same effect in the interlayer process. The ratio between the intra- and inter-layer edges is crucial here and will affect the results of modularity. This needs to be discussed, with references to detailed explanations (Bassett et al. 2013; Bazzi et al. 2016; Pilosof et al. 2016). Because the interlayer edge values were not measured empirically, there is no objectively correct value for ω . Hence, assigning a value of $\omega=1$ is not enough. The authors should explicitly measure modularity across a range of ω values. This was done for example in the original paper applying this approach (Mucha et al. 2010) and in (Bassett et al. 2013). For this, I would follow the logic and the Supp Info in (Bassett et al. 2011).

IMPORTANT: In the absence of empirical data on interlayer edge values, how to choose a correct value is an open question in network science in general, and in ecology in particular. Yet this should not be viewed as a weakness of the manuscript, just as it is not considered a weakness in other multilayer papers in the more general network science. On the contrary! Because it is a new application in ecology, this is what makes this paper novel. Exploring a range of ω informs us on how the relative importance of intralayer processes (namely, seed dispersal) and inter-layer processes (e.g., animal movement) affects the structure of ecological communities (see discussion in (Pilosof et al. 2016)). This is precisely the new insight and what makes this manuscript novel, as compared to previous studies that considered multiple but disconnected networks. Also see my comments below on analyzing aggregated networks.

6. Line 204 and on: No need to describe the Louvain algorithm here. Leave it to the SI. Instead, it is important to specify that modularity was calculated using an objective function and what this function quantifies. Actually, it is exactly like any application of the monolayer version of this function to monolayer networks (e.g., (Olesen et al. 2007; Fortuna et al. 2010; Thébaud 2013)), but extended to a multilayer approach. This will allow readers to also relate with what they already know from monolayer networks. Also, what is adapted for bipartite networks is not the general Louvain algorithm, but the objective modularity function Q . This is actually stated in the SI.

7. Modularity null model (lines 221-224). What is the hypothesis that the null model tests? Why did the authors choose this null model over others? What is being shuffled and what properties of the networks does this model conserve? I also recommend to have a null model that shuffles the interlayer edges. This will support the case that interlayer connectivity provides new insights.

8. Edge overlap: The interpretation of this network diagnostic needs more clarification. I did not understand how it gives us information about the contribution of each habitat to the overall process of seed dispersal. Also, "overall process of dispersal" is vague and I did not understand what it means.

9. Normalized degree and d' : How these properties support the aim of quantifying spatial coupling is not well explained. Was the goal of the calculations to see if they are affected by layer? If they add little information, I would either move that to the SI or not consider these measures. If this is kept in the manuscript then a much better interpretation and explanation of the logic behind these properties and their use is required. For d' it is better because it shows us that versatile animals are more generalists. This is good information but it is mentioned only in the results and it is not discussed. Also, a full model list along with AIC results should be given in the SI.

10. Versatility: If I am not wrong, versatility is not adapted to bipartite networks. In that case, versatility should be calculated on projections within each layer. That is, each layer is a network of animals in which two animals are connected if they disperse the same seed and the value of their interaction can be, for instance, the number of seeds shared or a measure of similarity. This is common in studies that apply centrality in bipartite networks in ecology. The ecological interpretation is important -- what is a versatile animal in such a multilayer network? Also, to clearly demonstrate the insights gained by simultaneously considering the different habitats, I would compare the versatility to the centrality of animals in an aggregated network (and please clearly define how you aggregated the layers), and in each layer alone, similarly to what was done in the original paper (De Domenico et al. 2015). I believe that this kind of analysis serves the aims more than the calculation of, e.g., normalized degree. Versatility has never been used in ecology, although this was suggested in a conservation aspect (Pilosof et al. 2016), and so it is a novelty of this paper but it should be demonstrated and interpreted clearly. Within each layer versatility (or centrality) tells you how important a species is in dispersing seeds in a habitat, while in a multilayer versatility tells you how important it is for the whole habitat. So you get insights into the additional value of the species for conservation.

RESULTS

1. Modularity and state nodes: In multilayer modularity the same physical node (e.g., elephant) can be classified into several modules. That is, each replicate of the elephant in a given habitat (the state node) can be assigned to a different module. For example, an elephant in Miombo is assigned to module #1 and an elephant in a forest to module #2. By looking at table 1, I see this does not happen and the authors also state that in line 291. This is very (but very) surprising, to the point I think the authors should revise their results. Happen or not, this point is not addressed properly in the results or discussion. If this does not happen it would be good to know why and what is the ecological interpretation. This is worth revising and discussing because the same species assigned to different modules in different habitats may be important for connecting habitats.

2. Edge overlap: the values in Table S2 should be written next to the edges in Figure 2. The figure is meaningless without them. The figures are beautiful, though. I like the habitat drawings!

3. Please explain why you could not correlate d' and versatility. It is possible to correlate d' and

versatility across species, no? This will show you that generalist species are spatial couplers. Not surprising, I think. I would also correlate, or at least discuss, the association between movement and versatility. Elephants can move great distances but ants no, so they will be more versatile. This is important because for the modularity you assume that all animals have the same effect on interlayer coupling (because you give all of them the same value of ω), but versatility does not use the interlayer edge information.

4. The results are focused on animals. While animals are attractive, they are only one half on the equation. I think that providing results for plants, and at the minimum their module affiliation and versatility is no less important. This can be placed in the SI, if the authors feel it diverts the focus.

6. I did not see results for multistrength. Did I miss it?

7. Figure 1: I did not understand the overall network. How did the authors aggregate layers into the overall network? There are plenty of ways to do so. E.g., did they recalculate the proportional interaction frequency? In any case, I am not sure that the overall network is necessary here because no calculation was made for it (but keep it if you perform calculations on the aggregate network). What do the colors represent? I would use colours to indicate the affiliation of state nodes (i.e., a species in a layer) to modules. That includes the plants.

DISCUSSION

1. What is common to the species that were assigned into a single module? For example, Mongoose and Helmeted guineafowl?

2. A discussion on what would be missed by analyzing multiple but disconnected layers or an aggregated network is in place. It will also be insightful to analyze modularity in the aggregated network to show that the module composition and versatility is different compared to when separating the layers. It will emphasize the authors' claim that "It is thus vital to consider the natural connections between habitats when analysing networks that extend across habitats within the same landscape." In practice, such analysis should be very easy to do. (and it should be done using the same algorithms, for proper comparison).

3. Line 396-399: Where is the support for that claim? Brings me back to the previous point...

4. What is the quantitative component of animal seed dispersal? (Line 388)

5. Line 399-400: what is this high tendency? Which network property showed that?

6. Line 413: Can that be a result of less sampling effort directed at birds? Are you sure this is a "true" pattern rather than an artifact of the sampling or analysis?

7. Lines 434-443. I totally agree with these claims but the authors have no support for them in

their results. To do this, they need to compare multilayer to aggregated and multilayer to each layer separately. While these sentences can be kept here, they need more references, but even better, a proper analysis. I would also recommend to down-tone a bit because, as I stated earlier, there are many studies that use multiple networks to compare network structure across space or time. I would strengthen in the conclusion that the importance is in explicitly connecting the layers (using interlayer edges). This is the source of new insights because this connectivity allows to simultaneously consider ecological processes within and between layers.

8. I suggest some discussion on the assumptions/limitations. For example, the issue of uniform interlayer edge weights or the lack of empirical data to quantify them.

REFERENCES

I did not scrutinize for errors, but here are some I picked on the way. The authors should revise the references to make sure that the claims are actually shown in the references they cite.

Line 62: ref 9 is inadequate. Better use 17

Line 85: 18 is incorrect. Should be 16,17,59

Line 204. Remove 18 and leave 45 at the end. Also add 27.

Ref 13 and 59 point to the same paper. 13 is incorrect. 59 is 2016.

Minor comments:

Line 33: habitat fidelity is not defined

Line 37: Versatility is not defined. Note that it is not a common network property even outside ecology...

lines 192-193: I would not mention temporal networks here (or in any other part of the manuscript, like the conclusion) because it is confusing. The manuscript is about spatial connectivity, better to focus on that.

Line 214: interlayer coupling is ω not γ .

L 273: what does most interaction mean? In each layer? Or overall?

Please report values of modularity (Q) for the shuffled networks.

L 318: Why was it impossible to test for that correlation?

L 329-332: This sentences is a bit awkward and needs rephrasing, as well as down-toning.

Line 341-342: Where in the results did you show low spatial fidelity by the modules? Just like in the intro, the spatial fidelity term is not defined properly, and somewhat confusing.

Table 1: The number of habitats where present is the number of habitats in which a module exists? This is not properly defined and I kind of had to guess it.

Bassett, D. S., M. A. Porter, N. F. Wymbs, S. T. Grafton, J. M. Carlson, and P. J. Mucha. 2013. Robust detection of dynamic community structure in networks. *Chaos* 23:013142.

Bassett, D. S., N. F. Wymbs, M. A. Porter, P. J. Mucha, J. M. Carlson, and S. T. Grafton. 2011. Dynamic reconfiguration of human brain networks during learning. *Proceedings of the National Academy of Sciences of the United States of America* 118:7641–7646.

Bazzi, M., M. A. Porter, S. Williams, M. McDonald, D. J. Fenn, and S. D. Howison. 2016. Community detection in temporal multilayer networks, with an application to correlation networks. *Multiscale modeling & simulation: a SIAM interdisciplinary journal* 14:1–41.

De Domenico, M., A. Solé-Ribalta, E. Omodei, S. Gómez, and A. Arenas. 2015. Ranking in interconnected multilayer networks reveals versatile nodes. *Nature communications* 6.

Fortuna, M. A., D. B. Stouffer, J. M. Olesen, P. Jordano, D. Mouillot, B. R. Krasnov, R. Poulin, and J. Bascompte. 2010. Nestedness versus modularity in ecological networks: two sides of the same coin? *The Journal of animal ecology* 79:811–817.

Kivelä, M., A. Arenas, M. Barthelemy, J. P. Gleeson, Y. Moreno, and M. A. Porter. 2014. Multilayer networks. *Journal of Complex Networks* 2:203–271.

Mucha, P. J., T. Richardson, K. Macon, M. A. Porter, and J.-P. Onnela. 2010. Community structure in time-dependent, multiscale, and multiplex networks. *Science* 328:876–878.

Olesen, J. M., J. Bascompte, Y. L. Dupont, and P. Jordano. 2007. The modularity of pollination networks. *Proceedings of the National Academy of Sciences of the United States of America* 104:19891–198916.

Pilosof, S., M. A. Porter, M. Pascual, and S. Kéfi. 2016. The Multilayer Nature of Ecological Networks.

Poisot, T., E. Canard, D. Mouillot, N. Mouquet, D. Gravel, and F. Jordan. 2012. The dissimilarity of species interaction networks. *Ecology letters* 15:1353–1361.

Thébault, E. 2013. Identifying compartments in presence-absence matrices and bipartite networks: insights into modularity measures. *Journal of biogeography* 40:759–768.

Trøjelsgaard, K., P. Jordano, D. W. Carstensen, and J. M. Olesen. 2015. Geographical variation in mutualistic networks: similarity, turnover and partner fidelity. *Proceedings of the Royal Society of London B: Biological Sciences* 282:20142925.

Reviewer #3 (Remarks to the Author):

This manuscript by Timoteo et al. uses a very large seed dispersal dataset from Mozambique to explore spatial patterns in seed dispersal networks. It uses multiplex network analysis tools, which have recently begun to surface in ecological studies. Despite the enviable dataset and rigorous analyses, I felt that the manuscript was largely descriptive and lacked any hypotheses. It seemed to lean heavily on the analyses rather than questions, and produced results that were very specific to the study system/region (see L.33-39 and L.424-429 for summaries of these results). Therefore, I regret that I cannot recommend that it be published in a journal of such high ranking.

The lack of novel questions is already apparent from the Introduction. Large sections of this are system specific (L.44-51, 106-108) or detail different metrics that can be generated for multilayer networks (L. 80-104), both of which would fit better in the methods. At the end of the first paragraph, where the authors should set up the problem and novelty of the study (L. 72-78), they instead confound the problem statement (a need to consider spatial connection among habitats) and a method ('multilayer networks'). It's not clear precisely what's the novel question here, and until this is stated, it isn't clear why multilayer network approaches are even necessary or the best way to answer it. Then it outlines system-specific ambitions (L.116-118): "Here, we took advantage of recent developments in multilayer networks to explore how seed-dispersal by all potential disperser guilds might be shaping habitat connectivity in Gorongosa." What are the actual questions, or is this just about understanding Gorongosa?

The closest thing to specific aims was L. 121-133, though both of these objectives were somewhat vague about what was novel. How do these questions expand on previous work on spatial coupling of seed-dispersal communities except by using a different method? (e.g. Garcia et al 2010 *Conserv. Biol.* DOI: 10.1111/j.1523-1739.2009.01440.x; Rodriguez-Perez et al 2014 *Func Ecol* DOI: 10.1111/1365-2435.12276; See also the review on spatial patterns of seed dispersal by Nathan & Muller-Landau 2000 *TREE* [http://dx.doi.org/10.1016/S0169-5347\(00\)01874-7](http://dx.doi.org/10.1016/S0169-5347(00)01874-7)).

Moreover, the first aim of evaluating the spatial fidelity of seed-disperser communities isn't possible with the study design used here, whereby habitat types aren't replicated. Modules within a single location could easily arise through rare species that were only sampled once (and happened to be in that location). This doesn't necessarily mean that the species (and their interactions) show high spatial fidelity, it only shows that they are rare. Indeed, the most abundant seed disperser (baboons) were also the main connectors of modules across habitats. Without replication of habitat types, and repeated identification of the same module structure in each replicate habitat, it's not possible to draw any meaningful inferences from these modularity patterns. All you can say is that some species are rare (so tend to be found in only one habitat), whereas others are common.

The lack of clarity about what, if anything, is novel in this manuscript continues into the Discussion. The statement (L.331) "Yet, we know nearly nothing about how these networks intermingle across large landscape mosaics." is simply untrue. Hagen et al (2012 *Adv Ecol Res* <http://dx.doi.org/10.1016/B978-0-12-396992-7.00002-2>) had an extensive review on spatial aspects of ecological networks. More recently and in this journal, Frost et al (2016 *Nature Comm* doi:10.1038/ncomms12644) quantified precisely how interaction networks mingle across habitats (with replication of these habitats across landscapes). Theoreticians like McCann, Loreau and others have explored the dynamic implications of this cross-habitat coupling. Sure, many studies focus on networks in a single habitat, but there is a growing body of both theoretical and empirical work that has explicitly examined cross-habitat linkages and the influence of landscape structure on networks. For a journal of this ranking, it's necessary to be clearer about precisely what is new here rather than simply saying "we know nearly nothing" about this broad topic (which we now know something about).

Specific comments:

L.114-115 "However, most studies on mutualistic networks focus on specific sets of species". What does this mean? This manuscript also focuses on a specific set of species.

L.223: r2dtable is not a bipartite function. It is contained within the stats package and applies Patefield's algorithm, which randomises matrices while holding row and column marginal totals constant. The null model in bipartite (swapweb) calls this r2dtable algorithm, but also maintains connectance constant. Thus swapweb is more conservative, as Patefield's algorithm (r2dtable) will almost certainly result in significant differences when used on quantitative network data. The reason is because the null model treats a species having one link with a weight of 8 as equally probable to having 8 links with a strength of 1. This generates many null webs with unrealistic degree distributions, and thus a high probability that any real-world network differs from the null-model-generated networks. The authors should clarify which algorithm they used and also explain what it constrains/randomises, rather than just giving an R function and expecting the reader to look it up.

L.245 spelling "multcomp" (no i)

Reply to reviewers' comments:

Reviewer #1 (Remarks to the Author):

This interesting study applies multilayer network analysis to an impressive seed-dispersal dataset collected over the span of an entire year in Gongorosa National Park, Mozambique, that supports diverse populations of animal dispersers, including insects, birds, and mammals. Two aspects of the study really stand out. The first is the dataset, which is unusual in terms of the breadth of interactions it encompasses (there was an effort to identify the interactions of the “entire” seed-dispersal community, rather than arbitrarily circumscribing the types of species investigated) and its geographic context (seed dispersal remains poorly studied in African savanna and woodland habitats relative to tropical forests, and especially Neotropical forests). A second noteworthy aspect of the study is the use of multilayer network analysis, which has not yet been widely applied in ecological contexts (see also Pilosof et al. 2015, ref. 13 in the manuscript). This approach enables researchers, at least in theory, to analyze the spatial and temporal structure and dynamics of complex networks (for example, to analyze the linkages between different habitats in a landscape, as was done here).

The authors identify 13 distinct “seed-dispersal [sub]communities” in a highly modular overall network structure, many of which occur in more than one of the four habitat types analyzed (grassland, transition forest, dry forest, miombo; their Table 1 and Fig. 1). This crossover of communities across habitat types leads to a large number of interactions and network hub species that are shared across habitat types (Fig. 2); although the diversity of plant species, animal species, and plant-animal interactions varied across these four habitat types, the mean normalised degree and specialization of dispersers did not (Fig. 3). However, disperser specialization and versatility varied considerably among disperser species (Fig. 4).

I very much appreciate the work done here and think that there is the potential here for an important contribution to the literature. However, I think some important work remains to be done to achieve this. The novelty of the study in terms of data and approach is fairly easy to identify, but I am struggling a bit to apprehend the novelty, robustness, significance of the biological insight. When boiled down, the biological conclusions summarized in the previous paragraph seem to be either (a) not particularly surprising and/or (b) interpreted and presented at a fairly superficial, phenomenological level without much attention to biological mechanism. Put another way, although I appreciate the value of looking at biological phenomena in new ways, it is not clear how our understanding of this problem has really been substantively advanced by this new way of looking at it. I would challenge the authors to explain how their results challenge conventional understanding of seed dispersal, if indeed they do.

For example, consider the 13 seed-dispersal communities in Table 1—a result that seems to me to be entirely phenomenological, with no robust (or even hypothesized) insight into the biological mechanisms that would generate these groupings. Community #6 consists of two viverrid carnivores, a small antelope, a tenebrionid beetle, and two pycnonotid birds, whereas community #8 comprises only porcupine and sable antelope. What do these groupings mean,

and how/why do they arise? Or are they rather an artifact of the methodological approach? I myself am at a loss to explain these groupings in light of the respective ecologies, diets, and behaviours of the species involved; it seems to me far more likely that they are curious artifacts of sampling and analytical approach than that they reflect something biologically “real” that totally transcends the apparent natural history of the system as understood by me and many others. If the authors can convincingly explain these results, then that would be very interesting indeed, but it’s not clear that the data are up to this task.

Reply: We thank the reviewer the comments on the overall results presented in the manuscript. The interlayer edge strength is a key to outcome of the modularity function, especially the ratio of the inter- to intra-layer strength is fundamental, and for that reason for the resulting modules, and the 13 communities could in a way be seen as hypothetical. These grouping reflect species that interact more often than expected with each other than with species outside their own module (e.g. Guimerà & Amaral 2005, DOI: 10.1088/1742-5468/2005/02/P02001; **lines 72-82** of this manuscript), in this case these animals were found to disperse the same plant species more often than expected. In the new version of this manuscript we provide a new analysis where we looked at how the strength of the connection between the habitats (interlayer strength) may influence the structure of the spatial network, and tried to reveal the mechanistic process that may have led to these communities

Similarly, the finding that the four habitat types share many interactions and hub species (Fig. 2) is not particularly surprising given that many species occur in multiple habitat types (Table 1, Fig. 1) and that the most important dispersers (e.g., baboons) are described as “ubiquitous.” Even here the authors’ mechanistic interpretation of the results is not entirely clear. One of the authors’ reasons for using multilayer network analysis is that it avoids researchers’ having to make arbitrary decisions about who/what to include in the network, but the delineation of habitat boundaries itself seems like an example of this kind of semi-arbitrary distinction. So, does the generally high “multilayer versatility” of dispersers across habitat types truly reveal something real about the “spatial cohesion of seed dispersal,” or does it simply reflect that the researchers have defined the “habitat borders” in an artificial way (or in a way that is incongruent with the actual biology of the system)? Might it be more unambiguously informative to consider space as a continuous quantitative variable, as opposed to these loosely defined categories?

Reply: True. The new focus of the paper goes exactly in that direction, i.e. that it is critical to properly assess to what extent habitats constraints animal behaviour movement or if they are observer perceptions with little reflection on how natural communities are structured. We agree that using space as a continuous quantitative variable could indeed be informative, and we acknowledge this as a potential limitation of our approach should be tackled in future works (**lines 323-340**). However, considering for example the ongoing landscape fragmentation, in which borders may be a real feature of the landscape (though it may depend on its permeability to each animal cross-border movements), thus “converting habitats into categories” (Frost et al 2016: DOI 10.1038/ncomms12644). We believe that our approach can make an important contribution to understand how ecological processes may be structured across these habitats. We added some of this information to discussion (**lines 277-293**).

And finally, what are we to make of the among-species variation in versatility and specialization (Fig. 4)? Is this a function of differences in the species' diets? Of differences in their foraging behaviour, mobility, and home range? Of differences in their relative abundance? I suspect the answer is "all of the above," but without knowing anything about the relative contribution of these different factors, it is difficult to derive any expectation about the extent to which these results will be generalizable. Can the authors' specify any testable predictions arising from their results that future studies could address to reveal whether the network properties revealed here are fundamental vs. idiosyncratic?

Reply: We also think it's "all of the above", i.e. a mixture of mobility patterns and diet preferences but we have no data to test the relative importance of each driver. We have added this important issue to the discussion (**lines 329-340**).

Data quality is another factor contributing to my uncertainty about the validity and significance of these results. As noted above, I think this is a very impressive dataset in many respects, and I commend and congratulate the authors for their ambitious attempt to characterize the entire seed-dispersal community (although remember that small mammals can be very important seed dispersers), for doing so across an entire year in multiple sites, and for confirming the identity of difficult-to-identify seed types using DNA barcodes. However, there are many factors that could still profoundly influence the results of this kind of analysis, which I do not think have been adequately acknowledged or accounted for. These include: (i) potential misidentification of dung types, because ranger IDs and field guides are unreliable for many species with similar-looking dung—perhaps dung be identified using DNA barcodes, similar to what was done with plants?;

Reply: We acknowledge that this can be a source of some level of misidentification of dung. Unfortunately, due to logistical constraints when could not keep these samples, thus we cannot perform the suggested analysis. However, Gorongosa rangers are very experienced field rangers with extensive knowledge of the fauna, and used to track animals in the field. Most of these rangers were in fact hunters for many years, during the war times, and which feed their families on wild game. Besides, in many situations the observers directly observed the dung produced by large herds (e.g. impalas) and collected it directly after it has been produced, so that their origin is confirmed. We have now added this information to **lines 390-394**.

(ii) bias towards large-seeded plant species, as small-seeded species are notoriously difficult to detect in the absence of complementary germination trials;

Reply: Indeed, many frugivory studies focus only on large seeds and ignore small ones. This is definitely not the case of our study. Here all samples were carefully scanned under a microscope and all seeds detected under 40X amplifications were extracted and identified. This includes many seeds smaller than 1mm (*Ficus ingens*). Although we acknowledge some potential bias against smaller seeds (probabilistically more likely to passed undetected), we are convinced that this bias is residual in this study. We added some of this information to the methods section, **line 398-400**.

(iii) sampling biases leading to undersampling of certain species (notably birds),

Reply: Thanks for bringing this up and giving us a chance to straighten this point. We were aware of the difficulty of sampling birds from the beginning and therefore we directed a large effort to this group. During each sampling round, in each habitat, we dedicated a mist-netting session starting at dawn that last 5hrs, and using 90metres of net. As a result of this effort we captured 379 birds of 96 species, and the majority produced droppings which we then analysed. However out of these 96-bird species, only 9 species produced droppings containing viable seeds, and that is the real reason why birds are not more common in the final network. Therefore, we believe that is a real effect (birds are not the main important dispersers in Gorongosa) and not a sampling artefact. This idea is also confirmed by the very few observations recorded during transects and that did not greatly contribute to increase the number of frugivorous interactions with birds. We have added some of this information in the methods section (**lines 394-397**) and discussion (**lines 325-328**).

and the related (iv) failure to account or correct for disperser species' relative abundances and movements. With respect to this last point, on lines 393-400, the authors argue that baboons, elephants, and civets are the "most important" species for "enhancing network connectivity at the habitat and landscape level," and that "this importance would have been missed" if not for the multilayer network approach. But whereas I have no doubt that baboons, elephants, and civets are important seed dispersers in this system, it is unclear both (a) to what extent they actually *couple* habitats, since the authors do not appear to have assessed the actual movement of seeds across habitat boundaries (although I believe that this is highly likely),

Reply: We do not have the actual movement of the animals that we have sampled, thus we cannot be sure where the seeds were picked up. We appreciate the comments by the reviewer. In the new manuscript, we have highlighted the critical need to obtain real estimates of inter-edge connectivity, and how previous works have attempted to include that type of information in (**lines 62-68, and 78-85** in introduction and **lines 263-267, and 270-279** in discussion), and hope that the new analysis that now includes, besides the multilayer analysis, a comparison to an aggregated network and a monolayer network scenarios, helps to clarify some of the questions raised.

and (b) to what extent their inferred importance, specialization, and versatility is simply a function of their being highly abundant relative to other species and therefore having the largest number of scat samples examined. Conversely, it seems statistically inevitable that species with a small number of samples will be inferred to have relatively low importance and versatility, and relatively high specialization, simply because a small number of samples cannot reveal a large number of interactions. The analysis of completeness (Table S1) for the community at large does not seem to account for this relative variation in sampling across different species (better would be if each species' interactions were assessed using rarefaction, but this would not be possible for poorly species represented by only one or a few samples). It seems to me that the conclusions derived from this kind of analysis are likely to be highly sensitive to variation in all of these factors, again contributing to uncertainty about the

biological interpretation of the statistical patterns being reported.

Reply: This is indeed an interesting point. We agree that dispersers abundance influence sample size and sample size can potentially influence the results in terms of species importance or versatility. This could be particularly the case if we had fully characterized the trophic niche width of abundant species but have only partially characterized that of rare species (due to poor sampling). In this study, we tried to sample the whole community as homogeneously as possible, giving the same chance to detect interactions from all guilds. Therefore, we don't see animal abundance/sample size as an artefact but as a real driver of animals' importance in their communities. It is very common in network studies that the most topologically important species are also the most abundant ones. Therefore, it is likely that rare species will have a low community effect even if they are individually important/effective. We see this as a strength of using quantitative matrices and not as a weakness (e.g. Vázquez et al 2005, DOI: 10.1111/j.1461-0248.2005.00810.x). However, this was not always case in our work as can be seen by the importance obtained for elephants in the present work and which are far from being abundant in Gorongosa, or the low importance of birds though they are abundant in Gorongosa, and that we caught a fair number of species and individuals. We believe that this importance is real and is a result of individual effect of each animal and may be affected by the abundance of the species. Some of this information was added to the discussion (**lines 304-328**).

We added species accumulation curves (also by suggestion of reviewer #2), for animal species and plants (**lines 409-414** in Methods section and **lines 142-144** in Results sections, and Supplementary Results **Table S1 and Figure S1**).

In summary, I respect the work done here and I see great potential for this dataset and this kind of analysis to generate new and influential insights into the structure of mutualistic networks. But I also think that the authors are obliged to do more to ground their analysis in the biology of the system, to acknowledge (and where possible quantitatively analyze) the potentially biasing sources of variation, and to specify what can be done to further test or refute the conclusions of this study in other systems. If we do not insist upon these criteria, then it will be impossible to gauge the extent to which fresh analytical approaches such as multilayer network analysis genuinely reveal previously unappreciated dimensions of these systems, or whether they instead generate spurious and artifactual patterns that muddle the picture and lead future investigators down blind alleys.

Reply: Thank you very much for your insightful comments that helped to improve the manuscript.

Reviewer #2 (Remarks to the Author):

In this paper the authors explore the structure of a bipartite animal-seed network across 4 different habitat types in the Gorongosa national park. They aim to identify cohesive groups (modules) of animals and seeds across habitats, as well as to characterize the role of animals as couplers of seed dispersal across habitats. The novelty of the manuscript lies in its multilayer network approach, which is used to quantify the structural patterns across the 4 habitats simultaneously, thereby enabling explicit consideration of the effect of habitat connectivity on the structure of the network. Overall, I find the approach novel and adequate for the question in hand. As discussed in the manuscript, this approach also has an applied value for conservation. The ecological application for multilayer networks presented here can also be of interest for non-ecologists because until now the field of multilayer networks has seen very little ecological applications. I also appreciate the tremendous effort put into data collection, which adds to the strength of this paper.

That said, the paper has some weak parts that need considerable improvement. In particular, the calculations of network diagnostics and/or their interpretation in ecological terms is incorrect or missing in some instances. I also think that the paper is not focused enough, and this obscures its novelty. The authors calculate many network diagnostics, and these are not properly discussed/interpreted, while the interpretation of modularity, which is key to this manuscript is not good enough. I think it is better to calculate few diagnostics that serve to make the point and discuss them properly (see specific comments below). In particular, I would lose some of the less informative diagnostics and instead calculate modularity and versatility in an aggregated network and in each layer separately to show that the module composition and versatility change as compared to a multilayer network. This will give much more insight into the ecology of the system: What knowledge do we gain by breaking the aggregated network into habitats? What is the role of species within each layer and in the whole system? This will also strengthen the validity and novelty of the approach.

My overall opinion is that the paper has a novel and valuable idea and approach but needs to improve on the methodology and ecological interpretation, and it also needs a better focus in the application of the network approach. With such improvements, the paper will be adequate for the journal. Below I detail my comments and ideas for improvements.

Good luck!

Reply: Thank you very much for your fair and grounded evaluation. We tried hard not to fall in the trap of calculating a myriad of senseless diagnosis/descriptors in the first version of our manuscript and we had a true ecological hypothesis behind all of them (we were not simply pattern fishing). Nevertheless, we agree that comparing a large number of descriptors might distract the reader's attention from the main point, and therefore we did an additional effort to increase the focus of the paper by dropping some metrics (normalized degree, shares hub-species, and multidegree).

We appreciated the suggestion of calculating the modularity and versatility of the whole aggregated dataset, and for the aggregated and individual habitat networks, which we believe greatly improved the present work.

Major comments

INTRODUCTION

1. The terms “community” and “community structure” have different meanings in ecology and network science. I would define these in the beginning and use community for the ecological context, and “modularity” for the network context. (e.g., lines 97 and 205 can be especially confusing).

Reply: We thank the reviewer for pointing this out, which indeed could be a source of confusion for the reader. We have made clearer the existing distinction between “community” in an ecological sense, i.e. the assemblage of species in a given habitat/location (**lines 38-40**), and “community” in a network context of modularity, i.e. the set of nodes/species interacting more tightly between them -modules, than to nodes/species outside their modules (**lines 73-75**).

2. Lines 60-70: While it is correct that most ecological studies use monolayer networks, there are many many studies that use multiple networks, including some that investigate spatial structure (e.g., Trøjelsgaard et al. 2015). While the use of multiple networks does not explicitly consider interlayer connectivity (which is the novelty here), it does acknowledge the fact that network structure can change across space. This point is worth to bear in mind and also should be mentioned in the introduction. In addition, I would make the strong statements more subtle, to not offend researchers who did use multiple networks. Also, I guess that by “merging” (line 62) the authors mean aggregate? A short discussion on what is gained by considering interlayer edges, as opposed to using multiple networks (e.g., by applying Poisot’s framework from (Poisot et al. 2012) or aggregating networks will make the case of the study stronger. For example, in line 70: the distribution of interactions across habitats has been studied using multiple networks and for this interlayer edges are unnecessary. Finally, the statement on artificial borders or comfort zone is, to begin with, too strong and should be expressed more subtly; and more importantly, an inherent problem of how we subjectively perceive ecological systems. There is no absolute truth here. In this study the authors also defined the borders between the habitats artificially while in practice there may be intermediate transition zones or other ways to define the habitats.

Reply: You’re absolutely right, and we by no means had the intention to downplay the important work that has been done by others. We have down-toned all of the strong wording used in the first version of the manuscript and also acknowledge the important steps in other papers that tried to explore the effect of habitat borders and spatial variability on community structure, and species interactions (**lines 63-68**).

Also, by “merging” we meant aggregate, and have clarified that (**line 45**).

3. Lines 80-95. A short description of the ecological interpretation of these network diagnostics

is in place. What is species role? What does multidegree (or its monolayer version degree) mean in ecology?

Reply: We have added in the methods section a brief ecological interpretation of the network diagnostics used, which was indeed missing (**lines 528-536**). In the new version of the manuscript, and following previous suggestion, we have dropped some network diagnostic and multidegree was one of those that were not included in the present version.

4. Objectives: what is 'spatial fidelity'? How does a modular structure serve as a proxy for spatial fidelity? For me, spatial fidelity of a species is intuitively simply the proportion of habitats in which it occurs out of possible ones. I do understand how multilayer modularity can inform on groups of species that span across landscape types. 2nd objective: what is a "mobile link function"? And I would change "landscape cohesion" to dispersal of seeds across different landscapes (I guess this is what the authors mean). I would also just use "habitats", as used later on than landscapes.

Reply: We used the concept of spatial fidelity in a similar way as the reviewer used regarding species, but applying it to the communities found by the modularity algorithm. A community/module would have higher fidelity if it is restricted to one, or very few habitats, thus those species would interact strongly only in those few habitats. However, given that the new analysis moved away from a specific inter-layer edge strength towards a range of strengths, we think that that specific objective would not fit as it was, and we have re-written the objectives given the new approach presented here.

Mobile links are those species that can promote the connectivity between habitats because they can actively move across the landscape. In the present case, the species that go across habitats and can transfer seeds between them, this being a mobile link function (Lundberg & Moberg 2003, DOI: 10.1007/s10021-002-0150-4), and in this way contributing to the resilience of ecosystems. We added some of these details to the manuscript, as it was not indeed very clear where the concept came from and it hasn't been introduced before (**lines 51-54**).

We followed the suggestion of the reviewer regarding the use of "landscape cohesion" and the substitution of the term "landscape" by "habitat".

METHODS

1. Experimental design: I suggest to use species accumulation curves to determine collection completeness. I could not find any mentioning of the plant species. These should be mentioned in a dedicated table along with information on their abundance.

Reply: We have now included species accumulation curves for animal and for plant species in supplementary information. We kept the original table with sampling completeness in each habitat that shows the fraction of the estimated species richness detected in our network, using the estimator Chao2 (**lines 409-414** in Methods section and **lines 142-144** in Results sections, and Supplementary Results **Table S1 and Figure S1**).

2. Data and code: I was missing the R and Matlab codes and data to replicate the findings. This includes the files fed into MuxViz. If someone wants to replicate the results or take the approach and implement it on other data this is crucial!

Reply: This was indeed missing. We provide figshare links to Matlab and R code (Code Availability section at the end of the Methods section). Data will available upon request (Data Availability section at the end of the Methods section), but we provide here a temporary figshare link for reviewers access: [Redacted].

3. Please provide the equation for how you calculated interaction frequency. It is not clear enough. Especially so because it is a composite of several collection methods.

Reply: This was indeed not very clear. We improved this description to clarify how interaction frequency was calculated (**lines 418-425**).

4. A clearer description of the multilayer network is needed. What does it mean that layers were coupled by common species? The authors should state explicitly what are the interlayer edges. Do interlayer edges connect each species to itself across layers ('diagonal coupling'; see (Kivelä et al. 2014))? Do they connect animals and seeds from different layers? Also, what do these interlayer edges mean? For example, are they a proxy for animal movement? This point is crucial because the whole novelty of this paper is that it considers interlayer connectivity. Readers should have a feeling of what this interconnectivity means. It is also important to mention that this interconnectivity is used to calculate some (e.g., modularity) but not all (e.g., versatility, edge overlap) network diagnostics. Also, it is important to point out that physical entities (i.e., animal and plant species) can appear in different layers, and that different replicates of the same species in different layers are called 'state nodes'.

Reply: We appreciate the comment from the reviewer, and added a more detail description of the multilayer network, and how the different concepts apply to our own case. We believe that it now clarifies the points raised by the reviewer (**lines 426-438**).

5. Interlayer edge weights (ω): The interpretation of the interlayer coupling is not correct. It is true that the "balance" between interlayer and intralayer edge values can greatly affect the results. However, it is incorrect that $\omega=1$ means equivalent intra- and inter-layer coupling (line 220). If this were the case, then each intralayer edge should have also existed as an interlayer edge with the same value, which is clearly not the case. Instead, the authors should state that because the values of the interlayer edges were not empirically measured, they assigned them a value ($\omega=1$), and that this value is uniform across all the interlayer edges. This assumes that each animal and seed has the same effect in the interlayer process. The ratio between the intra- and inter-layer edges is crucial here and will affect the results of modularity. This needs to be discussed, with references to detailed explanations (Bassett et al. 2013; Bazzi et al. 2016; Pilosof et al. 2016). Because the interlayer edge values were not measured empirically, there is no objectively correct value for ω . Hence, assigning a value of $\omega=1$ is not enough. The authors should explicitly measure modularity across a range of ω values. This was done for example in the original paper applying this approach (Mucha et al. 2010) and in (Bassett et al. 2013). For this, I would follow the logic and the Supp Info in (Bassett et al. 2011).

IMPORTANT: In the absence of empirical data on interlayer edge values, how to choose a correct value is an open question in network science in general, and in ecology in particular. Yet this should not be viewed as a weakness of the manuscript, just as it is not considered a weakness in other multilayer papers in the more general network science. On the contrary! Because it is a new application in ecology, this is what makes this paper novel. Exploring a range of ω informs us on how the relative importance of intralayer processes (namely, seed dispersal) and inter-layer processes (e.g., animal movement) affects the structure of ecological communities (see discussion in (Pilosof et al. 2016)). This is precisely the new insight and what makes this manuscript novel, as compared to previous studies that considered multiple but disconnected networks. Also see my comments below on analyzing aggregated networks.

Reply: We appreciate the correction from the reviewer, and acknowledge that the interpretation given in the text was not precise. We changed this to provide a correct interpretation for the interlayer coupling. We also followed the suggestion to rephrase the reason for why it is assigned the same value to all interlayer edges. We greatly thank the suggestion to expand the assessment of modularity across a range of values as we believe that it boosts the importance of the current manuscript, and have included a brief discussion of the issue of the ratio of the inter- to intra-layer edge strengths (**lines 86-89, and 261-279**).

6. Line 204 and on: No need to describe the Louvain algorithm here. Leave it to the SI. Instead, it is important to specify that modularity was calculated using an objective function and what this function quantifies. Actually, it is exactly like any application of the monolayer version of this function to monolayer networks (e.g., (Olesen et al. 2007; Fortuna et al. 2010; Thébault 2013)), but extended to a multilayer approach. This will allow readers to also relate with what they already know from monolayer networks. Also, what is adapted for bipartite networks is not the general Louvain algorithm, but the objective modularity function Q . This is actually stated in the SI.

Reply: We followed the suggestion of the reviewer and kept the description of the Louvain algorithm to a minimum and leaving most of it to SI. We also made clearer the point about the modularity function being adapted for bipartite networks, and not the general Louvain algorithm (**Line 456-457**).

7. Modularity null model (lines 221-224). What is the hypothesis that the null model tests? Why did the authors choose this null model over others? What is being shuffled and what properties of the networks does this model conserve? I also recommend to have a null model that shuffles the interlayer edges. This will support the case that interlayer connectivity provides new insights.

Reply: We appreciate the questions raised by the reviewer, and acknowledge that the propose of the null model, and that the description of what it does to the matrices was very brief. We also want to thank the suggestion for a null model to reshuffle interlayer connectivity, and have add it in the current analysis. In the new manuscript, we present two null models, as was also suggested by the reviewer, and made a succinct description but with enough information, we believe, to understand what is being tested and what it does to the matrices (**lines 476-**

485).

8. Edge overlap: The interpretation of this network diagnostic needs more clarification. I did not understand how it gives us information about the contribution of each habitat to the overall process of seed dispersal. Also, "overall process of dispersal" is vague and I did not understand what it means.

Reply: We acknowledge that our explanation of edge overlap, and its interpretation its outcome was not very clear, and thank the reviewer for pointing this out. We have now improved it and hope that it is now more understandable (**lines 217-221** in the results section, **328-333** in the discussion section, **and 522-524** in the methods section).

9. Normalized degree and d' : How these properties support the aim of quantifying spatial coupling is not well explained. Was the goal of the calculations to see if they are affected by layer? If they add little information, I would either move that to the SI or not consider these measures. If this is kept in the manuscript then a much better interpretation and explanation of the logic behind these properties and their use is required. For d' it is better because it shows us that versatile animals are more generalists. This is good information but it is mentioned only in the results and it is not discussed. Also, a full model list along with AIC results should be given in the SI.

Reply: We thank the reviewer for the suggestions. We have removed normalized degree in the new manuscript but have kept specialization.

10. Versatility: If I am not wrong, versatility is not adapted to bipartite networks. In that case, versatility should be calculated on projections within each layer. That is, each layer is a network of animals in which two animals are connected if they disperse the same seed and the value of their interaction can be, for instance, the number of seeds shared or a measure of similarity. This is common in studies that apply centrality in bipartite networks in ecology. The ecological interpretation is important -- what is a versatile animal in such a multilayer network? Also, to clearly demonstrate the insights gained by simultaneously considering the different habitats, I would compare the versatility to the centrality of animals in an aggregated network (and please clearly define how you aggregated the layers), and in each layer alone, similarly to what was done in the original paper (De Domenico et al. 2015). I believe that this kind of analysis serves the aims more than the calculation of, e.g., normalized degree. Versatility has never been used in ecology, although this was suggested in a conservation aspect (Pilosof et al. 2016), and so it is a novelty of this paper but it should be demonstrated and interpreted clearly. Within each layer versatility (or centrality) tells you how important a species is in dispersing seeds in a habitat, while in a multilayer versatility tells you how important it is for the whole habitat. So you get insights into the additional value of the species for conservation.

Reply: We appreciate the point raised by the reviewer regarding the calculation of versatility in bipartite network, and that it should be done not for the bipartite network itself but for the unipartite projections of the two parts of it separately. We have reviewed that part of our analysis to comply with that need, explained how we obtained the projected network and what it means in the context of this work (**lines 509-521**). We also thank for the suggestion to

calculate versatility in the aggregated network, which we included in the present manuscript (**lines 525-526**), and have explained how this aggregated network is obtained (**lines 473-475**).

RESULTS

1. Modularity and state nodes: In multilayer modularity the same physical node (e.g., elephant) can be classified into several modules. That is, each replicate of the elephant in a given habitat (the state node) can be assigned to a different module. For example, an elephant in Miombo is assigned to module #1 and an elephant in a forest to module #2. By looking at table 1, I see this does not happen and the authors also state that in line 291. This is very (but very) surprising, to the point I think the authors should revise their results. Happen or not, this point is not addressed properly in the results or discussion. If this does not happen it would be good to know why and what is the ecological interpretation. This is worth revising and discussing because the same species assigned to different modules in different habitats may be important for connecting habitats.

Reply: We thank the comments from the reviewer, and we agree with it. This was indeed surprising and we had double-checked our results. In the analysis presented in the reviewed manuscript that same result occurs when estimating modularity using the same inter-layer strength. However, for the very low inter-layer strengths there are indeed nodes/species that change their module affiliation in different layer/habitats (**Fig. 3 and Fig. S4**).

2. Edge overlap: the values in Table S2 should be written next to the edges in Figure 2. The figure is meaningless without them. The figures are beautiful, though. I like the habitat drawings!

Reply: We followed the reviewer's suggestion and added the values of edge overlap to the figure. For that reason, Table S2 became redundant, and we have removed it.

3. Please explain why you could not correlate d' and versatility. It is possible to correlate d' and versatility across species, no? This will show you that generalist species are spatial couplers. Not surprising, I think. I would also correlate, or at least discuss, the association between movement and versatility. Elephants can move great distances but ants no, so they will be more versatile. This is important because for the modularity you assume that all animals have the same effect on interlayer coupling (because you give all of them the same value of ω), but versatility does not use the interlayer edge information.

Reply: Thank you for your comments and suggestions. We have now added the correlation between versatility and average d' (**lines 222-224** in the results section, **and 524-527** in the methods section, **Fig 5A**). Unfortunately, we do not have data on the actual movement of the different species to calculate a correlation between versatility and movement of the different species, but discuss these relation between the versatility and mobility of the different species (**lines 334-337**).

4. The results are focused on animals. While animals are attractive, they are only one half on the equation. I think that providing results for plants, and at the minimum their module

affiliation and versatility is no less important. This can be placed in the SI, if the authors feel it diverts the focus.

Reply: We thank the suggestion from the reviewer and have added module affiliation for the in SI (**Fig. S2**).

6. I did not see results for multistrength. Did I miss it?

Reply: The results for multistrength were at the very end of the results section, where we present the correlation existing between versatility and multistrength (**lines 221-225**).

7. Figure 1: I did not understand the overall network. How did the authors aggregate layers into the overall network? There are plenty of ways to do so. E.g., did they recalculate the proportional interaction frequency? In any case, I am not sure that the overall network is necessary here because no calculation was made for it (but keep it if you perform calculations on the aggregate network). What do the colors represent? I would use colours to indicate the affiliation of state nodes (i.e., a species in a layer) to modules. That includes the plants.

Reply: The overall network pools all the interactions between all the species of animals and all species of plants across the four habitats, and corresponds to the aggregated network. The width of each species boxes is proportional to its frequency in the network, as is the width of the grey triangles that represent the links between the species on the levels (i.e. width of the link is proportional to the interaction frequency. This is a standard display of bipartite network when they are drawn using the function `plotweb()` from R package `bipartite`. We appreciate the suggestion to colour the nodes of the networks according to their module affiliation but we think that this is still an informative figure, as it gives the reader an idea of how diverse is the community of seed dispersers in terms of animal guilds, in the overall network and within each habitat. Also, we are adding schematic representations of the module affiliation of the different species in separate figures, illustrating their affiliation in the aggregated network, in each of the habitats, and also in the multilayer network with given a certain value of inter-layer strength (**Fig. 3**).

DISCUSSION

1. What is common to the species that were assigned into a single module? For example, Mongoose and Helmeted guineafowl?

Reply: Thank for pointing out that we had not explained properly what means to belong to the same modules. We have now corrected this, and explain that species in the same modules are interacting, thus dispersing the same plant species (**lines 279-281**), and also when defining modularity and module assignment (**lines 448-450**).

2. A discussion on what would be missed by analyzing multiple but disconnected layers or an aggregated network is in place. It will also be insightful to analyze modularity in the aggregated network to show that the module composition and versatility is different compared to when separating the layers. It will emphasize the authors' claim that "It is thus vital to consider the natural connections between habitats when analysing networks that extend across habitats within the same landscape." In practice, such analysis should be very easy to do. (and it should

be done using the same algorithms, for proper comparison).

Reply: We thank the reviewer the suggestion, which we followed, and we think that the new analysis provides support to that claim.

3. Line 396-399: Where is the support for that claim? Brings me back to the previous point...

Reply: We meant this as a general rule, that animals track resources across the landscape. But we hope that with the new analysis provided it is easier to see support on our own results for this claim.

4. What is the quantitative component of animal seed dispersal? (Line 388)

Reply: This is a common concept in seed dispersal studies, where seed dispersal effectiveness can be divided in a qualitative and a quantitative component (Schupp 2010), but none of these concepts are really relevant here, so we decided to change it for more common terms.

5. Line 399-400: what is this high tendency? Which network property showed that?

Reply: This comes from the results in the degree-degree correlation, in Figure 2b, and that showed the tendency of species to conserve their degree across the different layers of the network. In face of the new expanded analysis we have removed this metric from the manuscript.

6. Line 413: Can that be a result of less sampling effort directed at birds? Are you sure this is a “true” pattern rather than an artifact of the sampling or analysis?

Reply: This issue has also been raised by reviewer 1 and has been now improved and explained. We don't think there is a sampling biased against birds. We were aware of the difficulty of sampling birds from the beginning and therefore we directed a large effort to this group. During each sampling round, in each habitat, we dedicated a mist-netting session starting at dawn that last 5hrs, and using 90metres of net. As a result of this effort we captured 379 birds of 96 species, and the majority produced droppings which we then analysed. However out of these 96-bird species, only 9 species produced droppings containing viable seeds, and that is the real reason why birds are not more common in the final network. Therefore, we believe that is a real effect (birds are not the most important dispersers in Gorongosa) and not a sampling artefact. This idea is also confirmed by the very few observations recorded during transects and that did not greatly contribute to increase the number of frugivorous interactions with birds. We have added some of this information in discussion (**lines 324-326**).

7. Lines 434-443. I totally agree with these claims but the authors have no support for them in their results. To do this, they need to compare multilayer to aggregated and multilayer to each layer separately. While these sentences can be kept here, they need more references, but even better, a proper analysis. I would also recommend to down-tone a bit because, as I stated earlier, there are many studies that use multiple networks to compare network structure across space or time. I would strengthen in the conclusion that the importance is in explicitly

connecting the layers (using interlayer edges). This is the source of new insights because this connectivity allows to simultaneously consider ecological processes within and between layers.

Reply: Thanks. We agree and we believe that the new analyses conferrers much support for these claims.

8. I suggest some discussion on the assumptions/limitations. For example, the issue of uniform interlayer edge weights or the lack of empirical data to quantify them.

Reply: We appreciate the comment from the reviewer, and have included mentions to some of the limitations of the approach of the work presented here, and that indeed should be clear to the readers. (e.g. **lines 262-278**).

REFERENCES

I did not scrutinize for errors, but here are some I picked on the way. The authors should revise the references to make sure that the claims are actually shown in the references they cite.

Reply: We appreciate the attention to these details from the reviewer, and have corrected the inadequate references pointed, and carefully revised the references used.

Line 62: ref 9 is inadequate. Better use 17

Line 85: 18 is incorrect. Should be 16,17,59 ()

Line 204. Remove 18 and leave 45 at the end. Also add 27.

Ref 13 and 59 point to the same paper. 13 is incorrect. 59 is 2016.

Minor comments:

Line 33: habitat fidelity is not defined

Line 37: Versatility is not defined. Note that it is not a common network property even outside ecology...

Reply: Thank you for pointing out that. We have now added the definition for versatility, and our ecological interpretation for it (**lines 505-520**).

lines 192-193: I would not mention temporal networks here (or in any other part of the manuscript, like the conclusion) because it is confusing. The manuscript is about spatial connectivity, better to focus on that.

Reply: We appreciate the comment from the reviewer, and have removed any mention to temporal networks, here and elsewhere when not appropriated.

Line 214: interlayer coupling is ω not γ .

Reply: We did not understand the comment from the reviewer, but we reckon there must have been some confusion with the Greek symbols, as in the specified sentence ω (ω) is being used for inter-layer coupling.

L 273: what does most interaction mean? In each layer? Or overall?

Reply: We meant most interactions in the overall network. We have now specified it in the manuscript (**line 135**).

Please report values of modularity (Q) for the shuffled networks.

Reply: The modularity values of the shuffled networks are not given in the **Fig. 2 and Fig. S3**.

L 318: Why was it impossible to test for that correlation?

Reply: Indeed, it is possible to calculate this correlation and we have now added (**lines 22-222**).

L 329-332: This sentences is a bit awkward and needs rephrasing, as well as down-toning.

Reply: We have rephrased this section of the manuscript.

Line 341-342: Where in the results did you show low spatial fidelity by the modules? Just like in the intro, the spatial fidelity term is not defined properly, and somewhat confusing.

Reply: We agree that this was not very well explained. We have now dropped this objective and have rewritten this objective given the new approach taken. We have addressed this in the reply to the reviewer's comment #4 above regarding the "objectives" section.

Table 1: The number of habitats where present is the number of habitats in which a module exists? This is not properly defined and I kind of had to guess it.

Reply: Given the new analysis this table is not present anymore.

Bassett, D. S., M. A. Porter, N. F. Wymbs, S. T. Grafton, J. M. Carlson, and P. J. Mucha. 2013. Robust detection of dynamic community structure in networks. *Chaos* 23:013142.

Bassett, D. S., N. F. Wymbs, M. A. Porter, P. J. Mucha, J. M. Carlson, and S. T. Grafton. 2011. Dynamic reconfiguration of human brain networks during learning. *Proceedings of the National Academy of Sciences of the United States of America* 118:7641–7646.

Bazzi, M., M. A. Porter, S. Williams, M. McDonald, D. J. Fenn, and S. D. Howison. 2016. Community detection in temporal multilayer networks, with an application to correlation networks. *Multiscale modeling & simulation: a SIAM interdisciplinary journal* 14:1–41.

De Domenico, M., A. Solé-Ribalta, E. Omodei, S. Gómez, and A. Arenas. 2015. Ranking in interconnected multilayer networks reveals versatile nodes. *Nature communications* 6.

Fortuna, M. A., D. B. Stouffer, J. M. Olesen, P. Jordano, D. Mouillot, B. R. Krasnov, R. Poulin, and J. Bascompte. 2010. Nestedness versus modularity in ecological networks: two sides of the same coin? *The Journal of animal ecology* 79:811–817.

Kivelä, M., A. Arenas, M. Barthelemy, J. P. Gleeson, Y. Moreno, and M. A. Porter. 2014. Multilayer networks. *Journal of Complex Networks* 2:203–271.

Mucha, P. J., T. Richardson, K. Macon, M. A. Porter, and J.-P. Onnela. 2010. Community structure in time-dependent, multiscale, and multiplex networks. *Science* 328:876–878.

Olesen, J. M., J. Bascompte, Y. L. Dupont, and P. Jordano. 2007. The modularity of pollination networks. *Proceedings of the National Academy of Sciences of the United States of America* 104:19891–198916.

Pilosof, S., M. A. Porter, M. Pascual, and S. Kéfi. 2016. The Multilayer Nature of Ecological Networks.

Poisot, T., E. Canard, D. Mouillot, N. Mouquet, D. Gravel, and F. Jordan. 2012. The dissimilarity of species interaction networks. *Ecology letters* 15:1353–1361.

Thébault, E. 2013. Identifying compartments in presence-absence matrices and bipartite networks: insights into modularity measures. *Journal of biogeography* 40:759–768.

Trøjelsgaard, K., P. Jordano, D. W. Carstensen, and J. M. Olesen. 2015. Geographical variation in mutualistic networks: similarity, turnover and partner fidelity. *Proceedings of the Royal Society of London B: Biological Sciences* 282:20142925.

Reviewer #3 (Remarks to the Author):

This manuscript by Timoteo et al. uses a very large seed dispersal dataset from Mozambique to explore spatial patterns in seed dispersal networks. It uses multiplex network analysis tools, which have recently begun to surface in ecological studies. Despite the enviable dataset and rigorous analyses, I felt that the manuscript was largely descriptive and lacked any hypotheses. It seemed to lean heavily on the analyses rather than questions, and produced results that were very specific to the study system/region (see L.33-39 and L.424-429 for summaries of these results). Therefore, I regret that I cannot recommend that it be published in a journal of such high ranking.

The lack of novel questions is already apparent from the Introduction. Large sections of this are system specific (L.44-51, 106-108) or detail different metrics that can be generated for multilayer networks (L. 80-104), both of which would fit better in the methods. At the end of the first paragraph, where the authors should set up the problem and novelty of the study (L. 72-78), they instead confound the problem statement (a need to consider spatial connection among habitats) and a method ('multilayer networks'). It's not clear precisely what's the novel question here, and until this is stated, it isn't clear why multilayer network approaches are even necessary or the best way to answer it. Then it outlines system-specific ambitions (L.116-118): "Here, we took advantage of recent developments in multilayer networks to explore how seed-dispersal by all potential disperser guilds might be shaping habitat connectivity in Gorongosa." What are the actual questions, or is this just about understanding Gorongosa?

The closest thing to specific aims was L. 121-133, though both of these objectives were somewhat vague about what was novel. How do these questions expand on previous work on spatial coupling of seed-dispersal communities except by using a different method? (e.g. Garcia et al 2010 *Conserv. Biol.* DOI: 10.1111/j.1523-1739.2009.01440.x; Rodriguez-Perez et al 2014 *Func Ecol* DOI: 10.1111/1365-2435.12276; See also the review on spatial patterns of seed dispersal by Nathan & Muller-Landau 2000 *TREE* [http://dx.doi.org/10.1016/S0169-5347\(00\)01874-7](http://dx.doi.org/10.1016/S0169-5347(00)01874-7)).

Moreover, the first aim of evaluating the spatial fidelity of seed-disperser communities isn't possible with the study design used here, whereby habitat types aren't replicated. Modules within a single location could easily arise through rare species that were only sampled once (and happened to be in that location). This doesn't necessarily mean that the species (and their interactions) show high spatial fidelity, it only shows that they are rare. Indeed, the most abundant seed disperser (baboons) were also the main connectors of modules across habitats. Without replication of habitat types, and repeated identification of the same module structure in each replicate habitat, it's not possible to draw any meaningful inferences from these modularity patterns. All you can say is that some species are rare (so tend to be found in only one habitat), whereas others are common.

The lack of clarity about what, if anything, is novel in this manuscript continues into the Discussion. The statement (L.331) "Yet, we know nearly nothing about how these networks intermingle across large landscape mosaics." is simply untrue. Hagen et al (2012 *Adv Ecol*

Res <http://dx.doi.org/10.1016/B978-0-12-396992-7.00002-2>) had an extensive review on spatial aspects of ecological networks. More recently and in this journal, Frost et al (2016 Nature Comm doi:10.1038/ncomms12644) quantified precisely how interaction networks mingle across habitats (with replication of these habitats across landscapes). Theoreticians like McCann, Loreau and others have explored the dynamic implications of this cross-habitat coupling. Sure, many studies focus on networks in a single habitat, but there is a growing body of both theoretical and empirical work that has explicitly examined cross-habitat linkages and the influence of landscape structure on networks. For a journal of this ranking, it's necessary to be clearer about precisely what is new here rather than simply saying "we know nearly nothing" about this broad topic (which we now know something about).

Reply: Despite the mostly negative review, we don't totally disagree with this opinion, and we see where the disappointment of the reviewer comes from. Indeed, in the first version of the manuscript one could get the idea that the complex methodologies and analyses were not matched by ecological advances or new findings. Nevertheless, we are very happy now because the new analyses suggested by reviewers 1 and 2 come to solve this issue. We now use the same data to actually show the importance of quantifying explicitly inter-layer edge, and how this can lead to different conclusions about the structure of multilayer networks. We also incorporated many of these suggestions, regarding previous literature dealing with inter-habitat species mobility in the introduction (**lines 67-68**), and tried to make clear the difference between how previous studies have established habitat connectivity across a landscape and how we explicitly incorporate that connection in the network analysis itself. We hope that the reviewer will be much happier when he reencounters this paper.

Specific comments:

L.114-115 "However, most studies on mutualistic networks focus on specific sets of species". What does this mean? This manuscript also focuses on a specific set of species.

Reply: It is true that here we focus on a specific set of species, though interactions were not sampled with *a priori* group of species in mind (say, just birds, or just ungulates). We sampled all species we possibly could (**line 96-97**): "including all possible guilds of seed dispersing animals", but of course the subsequent analysis has to focus on the specific set of species that ended up in the network.

L.223: r2dtable is not a bipartite function. It is contained within the stats package and applies Patefield's algorithm, which randomises matrices while holding row and column marginal totals constant. The null model in bipartite (swapweb) calls this r2dtable algorithm, but also maintains connectance constant. Thus swapweb is more conservative, as Patefield's algorithm (r2dtable) will almost certainly result in significant differences when used on quantitative network data. The reason is because the null model treats a species having one link with a weight of 8 as equally probable to having 8 links with a strength of 1. This generates many null webs with unrealistic degree distributions, and thus a high probability that any real-world network differs from the null-model-generated networks. The authors should clarify which algorithm they used and also explain what it constrains/randomises, rather than just giving an R function and expecting the reader to look it up.

Reply: Thank you for pointing out the imprecision in the specification of the null model, and that was indeed not enough explained. In the new manuscript, we have explained exactly which null models are used and what is randomized (**lines 475-484**).

L.245 spelling "multcomp" (no i)

Reply: Thank you for pointing this typo out.

Reviewers' comments:

Reviewer #1 (Remarks to the Author):

I appreciate the work that the authors have done here. This is an impressive and important dataset that I think needs to be published. However, I am unable to recommend publication of the manuscript in its present form, for many of the same reasons put forward by the three original reviews. These reviews were rather voluminous in their constructive criticism -- especially the comments of Reviewer 2, which were comprehensive and outstanding -- and although the authors have taken positive steps forward to address some of the concerns, I think that they could have gone farther, and I really wish that they had been a bit more elaborate in explaining exactly what they had done in response to each concern (many of the responses simply state something to the effect of "We have addressed this, see lines XXX-YYY").

The biggest problem in my view is that the biological interpretation of the results is extremely difficult to parse, for any generally informed scientific reader and especially for a nonspecialist in the network literature, because they are presented in dense jargon. This was critiqued at length by previous Reviewer 2, who although clearly an expert on networks, nonetheless found the presentation opaque. Some steps have been taken to remediate this issue, mainly in the Methods and the Supporting Information, but the flavour of the interpretation still needs to come through (much) more clearly in the main text for this study to be accessible to a broad audience of biologists. For example, the entire Results section from lines 146 to 197 is couched almost entirely in specialized jargon, with no hints as to the biological interpretation and intuition underlying the findings. I realize that journals place limits on word counts, but this does not absolve the authors of the responsibility of presenting the results in a clear way.

The two main strengths of this study are (1) the impressive dataset that has been assembled, which the authors correctly argue is a first for this kind of ecosystem, and (2) the application of a form of network analysis that is still novel in the ecological literature. However, the previous Reviewer 3's criticism about the lack of conceptually motivated questions and hypotheses does not to me seem to have been satisfactorily resolved. The authors still rest their case on the novelty of the approach and the dataset, which to my mind is only half the battle. I do not think it is imperative for the authors to demonstrate that their results are "general" in the sense of applying to many other systems; but this is where clear and conceptually motivated guiding questions and hypotheses would help us better understand what insights from this study we could conceivably carry forward to test in other systems. I also do not feel that the authors have even clinched the case that we now have a much better understanding even of this particular system, due in part to the murkiness of the presentation. (Figs. 3 and S2 are extremely difficult to assimilate; can a tabular version be presented somewhere showing which species belong to which modules?)

It also is not clear to me that the multilayer approach has yielded substantially richer or more robust insights from what a monolayer approach would have told us. The authors have attempted to show this via statistical comparison of the multilayer and non-multilayer approaches, showing that this can lead to different inferences about the membership of species in different modules, and that the multilayer approach gives us information about the linkages between different habitat types, but again, I am left grasping as to how to articulate what that really means, in a general biological sense. I think it is incumbent upon the authors to do that for us. I wish that they would put less effort towards proclaiming the value of the study and the approach and more towards actually demonstrating the value of the study and approach via a lucid and forthright presentation.

The issue of artifacts in the interpretation was raised by all three previous reviewers, and I am not fully persuaded that it has been satisfactorily addressed. The authors mention several times that

many prior network studies are artificially circumscribed by habitat boundaries, implying that their approach has greater fidelity to the biology of the system by spanning multiple habitat types -- but as noted by previous reviewers 1 and 2, the very notion of habitat types is itself an abstract and subjectively defined criterion. In that light, the logic of the authors motivating rationale seems to falter.

In a related vein, the authors' have responded to criticisms about artifacts stemming from differences in species' relative abundance by including species accumulation curves and also by arguing that they have faithfully sampled organisms in relation to their relative abundance on the landscape which should reflect their overall net importance to seed dispersal function; this is somewhat reassuring, but I think it underestimates the inevitable sampling biases that arise in the course of work like this. I still feel that an approach based on rarefied data, or some other way of accounting for the potential skew in inferences arising from the fact that some species are very rare (or at least very rarely sampled) and others are common (or commonly sampled), would be most appropriate for illuminating the true architecture of the system. Or at least for giving a quantitative indication of how different or not the results would be if the data were controlled in such a way.

Specific comments:

* Reviewer 2 requested that the data be published; the authors responded that they will make data available on request. I personally feel it is imperative for a study like this to make the data publicly available on publication, either as a supplementary file to this study or via a companion publication on Dryad or other repository. Not least because that will greatly increase the impact and citation frequency of this study by enabling others to apply the data to related questions. This will not cost the authors anything, and it would be a great benefit to the community.

* Reviewer 2 asked for a full model list with AIC results; I do not see that this has been provided.

* On line 236, there is a strange statement about the study ecosystem being "one of the most diverse landscapes in the World"; I don't really know what this means, but in any event it seems like somebody's opinion, not something that has been (or can be) objectively evaluated.

In conclusion, I believe that there is potential in this study -- certainly in the dataset, perhaps in the multilayer network approach as applied to the data -- and I applaud the extensive work that the authors have done, but I think that they still need to push themselves harder in order to bring out the real significance and meaning of their study for the broad audience that they are aspiring to reach. The argument that this is a novel approach in the context of the ecological literature is not in my view sufficient in the absence of a more convincing set of arguments about how our understanding of the underlying biology has been advanced, deepened, diversified, etc. I think that many of my lingering concerns could be addressed with an overhaul of the presentation, by developing lines of argument that are currently underdeveloped, and by backing off a few lines of argument that remain stronger than the data warrant, but this will take more than cosmetic edits.

Reviewer #2 (Remarks to the Author):

In this reviewed version I find the authors did a great job in incorporating the suggestions from the first review. Overall, I find this version more ecologically sound and better on the interpretation. I still have some issues but I believe most of my comments are a matter of rewriting parts of the manuscript.

I do not wish to delay the publication of this manuscript any longer. Therefore, it is ok for me to not have a third round, provided that the authors do address all my comments to the satisfaction of the editors.

NOTE: this was an easier version to review and my comments were more specific so I incorporated them directly in the PDF. Please go through them carefully. I also suggest to read all of them first because some of them inter-relate although they are in different parts of the manuscript.

Thank you for the opportunity to review the paper and I look forward to seeing that published.

Reviewer #3 (Remarks to the Author):

The authors have significantly revised the manuscript. My main concern on the original version was that the questions and the applicability of results beyond this specific system were unclear, and did not do justice to the detailed dataset. I think this aspect has been much improved, with specific questions being clarified, previous work on cross-habitat linkages being better recognised, and a greater focus on concepts rather than methodology in the introduction.

I think though, that this focus could be clearer in the abstract (e.g. L.30-33). Much of this will mean nothing to readers unless they read the whole paper (e.g. the abstract doesn't even say what the layers are, so interlayer connectivity is meaningless). I'd focus the abstract more around the main questions of the paper (L. 110-124): i.e. how do multilayer networks with spatial connections among habitats (i.e. layers) differ from either the single habitats or the habitats just aggregated together? Also, what does a disperser's versatility tell you that traditional species-level metrics don't? I'd give precise answers to these questions in the abstract.

I am left with only one major concern about the ms. L.418 Interaction frequencies were calculated by pooling together data that were sampled in very different ways, and I am concerned that this could generate some biases. For example, the probability of observing dung along a transect will correlate with the size of the dung, and I expect that this was the reason that the authors used different approaches (mist netting etc) to identify dispersal by birds. However, because the analysis compares species with each other, it is absolutely necessary that sampling is equivalent across species. Otherwise, some species could appear to be less important as connectors simply because they are under-sampled. I'm not sure what the retention time of dung would be (before decomposition or removal by insects), but I imagine that dung could persist over several days (or even weeks), and will be more likely to do so if it is large (i.e. comes from a large animal). This means that transects would capture a large number of dispersal events by even a single individual large mammal, whereas mist nets only capture a single fecal sample from each bird. Consequently, I would expect interactions by birds to be undersampled relative to interactions by animals (this suspicion was confirmed on L.127), and therefore I don't believe that the interaction frequencies are comparable (i.e. on the same scale) for species sampled using different techniques (birds vs. mammals). For the mixed models that conduct analyses within species, it would appear that this doesn't matter. However, if some plant species have a bird-dispersed strategy, they will appear to be less important and birds will appear to be less central than they should be (L.211-212 confirms that the species with highest versatility are mammals, and the conclusion on L.242-246 could be an artefact of many species' interlayer connectivity being poorly sampled). I could see two potential solutions to this problem: 1) the networks could be treated as unweighted, or 2) the sampling completeness analysis could be conducted separately for each method of sampling, to demonstrate that interactions (in addition to species) were sampled equivalently using the different methods, such that the frequencies captured by each method are directly comparable as link weights within a single network.

My remaining comments are all very minor:

L.24 "and static" may be true in the majority of cases, but there are still quite a large number of exceptions to this, so I'd delete it.

L.70, I think the names of the authors for ref 14 are missing (or at least some noun is needed where the ref number is).

L.84 grammar "nodes that can change module". Perhaps modules plural, and change how? Or do you mean that can occupy different modules in different layers?

L.91 change to "considerably".

The paragraph beginning L.90 doesn't work very well. First it says that species' roles are conserved, and then that centrality is often used to measure species roles. This implies that the references for role conservatism in the first sentence used centrality, and that's not the case (at least not in the Stouffer et al one, with which I'm most familiar). So I think the justification for focusing on centrality needs to be better defended. I'm not even sure it's needed here, given that only modularity is mentioned in the specific questions below.

L.119, L.199 "disperser" singular for adjectives

L.167-9 "The structure...being significantly lower" How can structure be lower? Do you mean modularity?

L.172 add ", which" after modules

L.229 "habitat borders"

L.238 "understanding"

L.246 spelling "ensuring"

L.252-3, words repeat

L.261-263 Doesn't this contradict the result on L.172-173 that the aggregated network was in line with the multilayer network? I would explain this result in laypersons terms, as it's the first main objective of this paper to compare the two approaches. Something along the lines of "This means that species may appear to interact with others in different parts of the network (i.e. across modules), but they tend to do this only across certain habitats, such that species tend to interact with subsets of species within subsets of spatially-coupled habitats". This probably isn't exactly right, but an explanation in simple terms like this would be useful.

L.264 "empirically"

L.279 across what? (noun missing).

L.403 change "was" to "were"

L.468-469 The interlayer strength was the same for all species. I take it that this was the same for all species that actually had an interlayer link (i.e. occurred in two habitats), not all species in the aggregated network? Perhaps add a couple of words to clarify this.

L.434 The authors use occurrence in multiple habitats as a proxy for movement. The caveat should be made that occurrence across habitats doesn't equal frequent dispersal across habitats, which is what actually matters for cross-habitat connectivity. I realise you don't have another option here, but it's still an unproven assumption.

L.474 "layers" plural

L. 497 The use of d' as a specialization measure is increasingly criticised, in part because of its low informativity (signal to noise ratio, see Poisot et al. 2012), and in part because it measures selectivity within a range of resources, rather than the range itself, so its interpretation can be counterintuitive when compared with other measures like degree. If you're going to use only this metric, it should be defended.

L.502 spelling "Akaike"

L.505 "Residuals were inspected for departures from normality". Why? I thought you were assuming gamma, not normal errors (L. 498).

L.547 I think nowadays "data available upon request" isn't enough. They should be made available in a repository (or provided as supplementary material since this journal isn't paywalled).

L.564 "led", not "lead"

Reference

Poisot T, Canard E, Mouquet N, Hochberg ME. A comparative study of ecological specialization estimators. *Methods in Ecology and Evolution*. 2012 1;3(3):537-44.

Reviewer #1 (Remarks to the Author):

I appreciate the work that the authors have done here. This is an impressive and important dataset that I think needs to be published. However, I am unable to recommend publication of the manuscript in its present form, for many of the same reasons put forward by the three original reviews. These reviews were rather voluminous in their constructive criticism -- especially the comments of Reviewer 2, which were comprehensive and outstanding -- and although the authors have taken positive steps forward to address some of the concerns, I think that they could have gone farther, and I really wish that they had been a bit more elaborate in explaining exactly what they had done in response to each concern (many of the responses simply state something to the effect of "We have addressed this, see lines XXX-YYY").

Reply: Firstly, we would like to thank the insightful comments and relevant issues raised by the reviewer, which we address below hopefully to a satisfactory level, and that we think make the manuscript clearer. Also, and in addressing the reviewer's critique about the way we structured our previous replies, we now try to provide a more detailed explanation of the steps taken to address all reviewers' concerns, and not only pointing what we are adding/removing/editing in the MS. Was not our intention to look "lazy" in our replies.

- The biggest problem in my view is that the biological interpretation of the results is extremely difficult to parse, for any generally informed scientific reader and especially for a nonspecialist in the network literature, because they are presented in dense jargon. This was critiqued at length by previous Reviewer 2, who although clearly an expert on networks, nonetheless found the presentation opaque. Some steps have been taken to remediate this issue, mainly in the Methods and the Supporting Information, but the flavour of the interpretation still needs to come through (much) more clearly in the main text for this study to be accessible to a broad audience of biologists. For example, the entire Results section from lines 146 to 197 is couched almost entirely in specialized jargon, with no hints as to the biological interpretation and intuition underlying the findings. I realize that journals place limits on word counts, but this does not absolve the authors of the responsibility of presenting the results in a clear way.

- The two main strengths of this study are (1) the impressive dataset that has been assembled, which the authors correctly argue is a first for this kind of ecosystem, and (2) the application of a form of network analysis that is still novel in the ecological literature. However, the previous Reviewer 3's criticism about the lack of conceptually motivated questions and hypotheses does not to me seem to have been satisfactorily resolved. The authors still rest their case on the novelty of the approach and the dataset, which to my mind is only half the battle. I do not think it is imperative for the authors to demonstrate that their results are "general" in the sense of applying to many other systems; but this is where clear and conceptually motivated guiding questions

and hypotheses would help us better understand what insights from this study we could conceivably carry forward to test in other systems. I also do not feel that the authors have even clinched the case that we now have a much better understanding even of this particular system, due in part to the murkiness of the presentation. (Figs. 3 and S2 are extremely difficult to assimilate; can a tabular version be presented somewhere showing which species belong to which modules?)

Reply: Thank you for your comment. Indeed, we agree that in order to clearly explain the technicality of the analysis we might have neglected the clarity of the biological interpretation of the main findings, as highlighted by the three reviewers. This was partly because of the way we initially structured the manuscript, with the interpretation of important concepts being “hiding” in the Methods and Supplementary Information, which is not what readers read first. We have now re-structured the main text, especially the Results section, by presenting the results together with a clear biological/ecological interpretation of all technical concepts, thus making the jargon (only the extremely necessary) much easier to interpret for general readers. Regarding the Figs 3 and S2, we appreciate that those are not the simplest of the figures, but in our view, replacing them by tables, eventually assigning a number (instead of a colour) to each module would not make them easier to read. We believe that while the interpretation of these figures might not be initially intuitive, after reading the caption the figures becomes extremely easy to interpret and highly informative as they allow the reader to track how the composition of the identified communities change with the different approaches.

- It also is not clear to me that the multilayer approach has yielded substantially richer or more robust insights from what a monolayer approach would have told us. The authors have attempted to show this via statistical comparison of the multilayer and non-multilayer approaches, showing that this can lead to different inferences about the membership of species in different modules, and that the multilayer approach gives us information about the linkages between different habitat types, but again, I am left grasping as to how to articulate what that really means, in a general biological sense. I think it is incumbent upon the authors to do that for us. I wish that they would put less effort towards proclaiming the value of the study and the approach and more towards actually demonstrating the value of the study and approach via a lucid and forthright presentation.

Reply: Thank you for your comment. It was not our intention to rely only on the novelty of the methods and proclaim without justification the value of this work. We acknowledge that we might not have been totally successful in explaining the practical interpretation of the results, blurring what are the main scientific/ecological advances of this work. In this version of the manuscript we have worked tirelessly on this aspect, in order to clearly highlight the added value of implementing the multilayer approach in ecology. In a nutshell, by explicitly reflecting the way that species interact in real complex landscapes, this approach allows us to detect patterns that are not detected

by the traditional analysis based on either merged or split habitats. By doing this, not only this approach allows us a much deeper understanding of how ecological processes (including, dispersal, but many others) work across habitats, but maybe most importantly, it exempts researchers from making this remarkably important, and yet often ungrounded decision of where to place or omit habitat borders in complex landscapes (**lines 31-37** in the Abstract, **lines 187-189** and **lines 219-227** in the Results, and **lines 354-384** in the Discussion).

- The issue of artifacts in the interpretation was raised by all three previous reviewers, and I am not fully persuaded that it has been satisfactorily addressed. The authors mention several times that many prior network studies are artificially circumscribed by habitat boundaries, implying that their approach has greater fidelity to the biology of the system by spanning multiple habitat types -- but as noted by previous reviewers 1 and 2, the very notion of habitat types is itself an abstract and subjectively defined criterion. In that light, the logic of the authors motivating rationale seems to falter.

Reply: This is a good and important comment. It is true that we also had the necessity to define habitat units, and that there is always some subjectivity in defining them. We now tried to highlight more clearly that this problem of defining the best “working” units is actually one of the major improvements of the current approach as by explicitly incorporating habitat units in the study design and exploring how they are connected we can infer from the data (rather than relying on a priori decisions of the researchers) how much informative value (i.e. how much sense) do those habitats units hold for the particular research question. In the future, studies that follow this approach explicitly quantify inter-habitat connectivity when analysing the broader network structure, will be able to explore the putative independence of the habitats/layers without having to make the methodological decision of whether aggregating or splitting the layers (habitats).

- In a related vein, the authors' have responded to criticisms about artifacts stemming from differences in species' relative abundance by including species accumulation curves and also by arguing that they have faithfully sampled organisms in relation to their relative abundance on the landscape which should reflect their overall net importance to seed dispersal function; this is somewhat reassuring, but I think it underestimates the inevitable sampling biases that arise in the course of work like this. I still feel that an approach based on rarefied data, or some other way of accounting for the potential skew in inferences arising from the fact that some species are very rare (or at least very rarely sampled) and others are common (or commonly sampled), would be most appropriate for illuminating the true architecture of the system. Or at least for giving a quantitative indication of how different or not the results would be if the data were controlled in such a way.

Reply: Thank you for bringing up the important issue of sampling completeness. This is actually largely overlapping with the only main concern of the reviewer #3, and we provide a joint response under the first comment of reviewer 3, giving account of the action took to minimize this potential bias (please see below).

Specific comments:

- Reviewer 2 requested that the data be published; the authors responded that they will make data available on request. I personally feel it is imperative for a study like this to make the data publicly available on publication, either as a supplementary file to this study or via a companion publication on Dryad or other repository. Not least because that will greatly increase the impact and citation frequency of this study by enabling others to apply the data to related questions. This will not cost the authors anything, and it would be a great benefit to the community.

Reply: We agree with much of the analysis and have discussed and agreed on this issue with the editor during the first revision. In the long run, we want and will share the full data, but there are some important constraints preventing us from doing so now.

* Reviewer 2 asked for a full model list with AIC results; I do not see that this has been provided.

Reply: This was provided in the Supplementary Table S3. In this model, we only have one explanatory variable (habitat) against which we test animal specialization, and that is our full model. The reduced model is then obtained by removing the "habitat" variable: an intercept-only model (**lines 663-665**). We present the AIC for both cases in the same table (AIC_{reduced} = - 16.02; AIC_{model} = -12.36). Perhaps some of this information would have made clearer what the full and the reduced models were, and should have been included in the legend of table S3. We have now done it (**Supplementary Table S3 in Supporting Information**).

- On line 236, there is a strange statement about the study ecosystem being "one of the most diverse landscapes in the World"; I don't really know what this means, but in any event it seems like somebody's opinion, not something that has been (or can be) objectively evaluated.

Reply: Thank you for your comment. This statement is presented and supported in a science communication book by Prof. Edward O. Wilson, but we agree that it is more of an informed opinion than proved scientific evidence and therefore we agreed to remove it. That sentence now reads "We made use of a very comprehensive dataset collected in a highly diverse African landscape, including all potential disperser guilds." (**lines 324-326**).

In conclusion, I believe that there is potential in this study -- certainly in the dataset, perhaps in the multilayer network approach as applied to the data -- and I applaud the extensive work that the authors have done, but I think that they still need to push themselves harder in order to bring out the real significance and meaning of their study for the broad audience that they are aspiring to reach. The argument that this is a novel approach in the context of the ecological literature is not in my view sufficient in the absence of a more convincing set of arguments about how our understanding of the underlying biology has been advanced, deepened, diversified, etc. I think that many of my lingering concerns could be addressed with an overhaul of the presentation, by developing lines of argument that are currently underdeveloped, and by backing off a few lines of argument that remain stronger than the data warrant, but this will take more than cosmetic edits.

Reply: We indeed push ourselves harder to convince the reviewer and all readers of the concrete value of this new approach and how it can advance the whole field of ecology. This was certainly much more than a simple cosmetic edition and we are confident that the value of this contribution is clear now.

Reviewer #2 (Remarks to the Author):

In this reviewed version I find the authors did a great job in incorporating the suggestions from the first review. Overall, I find this version more ecologically sound and better on the interpretation. I still have some issues but I believe most of my comments are a matter of rewriting parts of the manuscript.

I do not wish to delay the publication of this manuscript any longer. Therefore, it is ok for me to not have a third round, provided that the authors do address all my comments to the satisfaction of the editors.

NOTE: this was an easier version to review and my comments were more specific so I incorporated them directly in the PDF. Please go through them carefully. I also suggest to read all of them first because some of them inter-relate although they are in different parts of the manuscript.

Thank you for the opportunity to review the paper and I look forward to seeing that published.

Reply: We would like to thank the insightful and very helpful comments from the reviewer, and we have re-written several parts of the manuscript to address the questions raised, and make the MS easier to all readers.

INTRO

L46-48: I suggest to be careful here because you also artificially define borders.

Reply: Thanks for pointing this out. Yes indeed, we are aware that by defining habitats we are also in a way defining artificial borders, although we did not established plots (which borders are usually artificial) but we worked with natural habitat borders that can be very easily perceived in the great rift valley context (for example the border between grasslands and transition forest, or between these and Mixed dry forest are readily observed from satellite imagery). We meant here that the potential connection between study sites (whether plots or habitats) is usually ignored, and each site is considered independent and discrete entities. We have re-written this sentence to make it clearer to the reader, and now it reads “To date, most studies have considered networks as entities with discrete borders defined by the experiment or sampling design, ignoring the potential across-border connections” (**lines 50-52**). We also would like to note that further down we acknowledge some of the efforts made to incorporate inter-habitat connections across these (more or less) artificial borders (**lines 70-78**).

L63-64: Which natural processes?

Reply: We added now a couple of examples of a natural process that can be better understood by considering the role of species as ecosystem couplers. The text now reads “, ignoring the role of different species as spatial couplers of ecosystems

may hinder our understanding of natural processes, e.g. the flux of energy, or nutrients, between aquatic and terrestrial systems, pollen transfer by insects across the landscape, or the dispersal of seeds of invasive species by birds¹⁴” (lines 66-70).

L75: Actually, mutualistic networks are commonly nested. Modularity is a structural feature of antagonistic networks like food webs or host-parasite networks. See Fontaine et al 2011 Ecol Lett.

Reply: Thank you for your comment. We did not want to say that all mutualistic networks are significantly modular, but instead that Modularity (lower or higher) rapidly become recognized as an informative network attribute that has been explored in many mutualistic (and also antagonistic) networks. While antagonistic networks might tend to show a greater modularity, several mutualistic networks also exhibit a modular pattern, such as in pollination networks (e.g. Olesen, J et al (2007), DOI: 10.1073/pnas.0706375104), or seed dispersal (Nogales, M et al (2016), DOI:10.1111/geb.12315, or Donatti, C. et al (2011), DOI: 10.1111/j.1461-0248.2011.01639.x), and general patterns are still hard to establish. We have rewritten the sentence, which now reads “A key structural pattern often described in most recent ecological networks is modularity (...)” (line 83).

RESULTS

In the results, the authors are "jumping into" reporting results without really explaining the different metrics they used AND the logic behind them. For example, why look at edge overlap? what new insight does this give us? These metrics are new to ecology and for the non-multilayer network ecology specialist (which is 99.9% of ecologists) it would be very difficult to follow and to extract meaningful ecological insights. I am aware that it is challenging to write results in a story-like manner that would encompass methods in some detail but not over-detailed, and where ecological interpretations are needed but leave something for discussion, etc etc. Especially with the word limits. But without that kind of writing, the message of the paper will not pass smoothly. Being my second time reading this, I get it. But would someone who has limited time and 20 other papers to read get the message instantly? I suggest to maybe work with the Editors on that issue?

Reply: We fully accept that some parts of the manuscript were not very accessible in the last version, in part due to the structure of the manuscript, and the reader would have to do quite a lot of jumping back and forth from results/discussion and the methods in order to gather the meaning of some of the results, especially those related to new techniques, or at least not very common in ecology. We have made a strong effort in rewriting the results section so that they are now presented in a “semi-digested” way. We now explain in an integrated and technically light manner why and how we performed each test, and simultaneously provide a brief ecological

interpretation of the key results. We are very happy with the new version of the results (thanks for your help).

L146: Here, there is a huge jump from intro to results. Basically, this is a matter of editing such that the reader would not have to go to the methods to understand this. See my previous point. Specifically for this part, the essential is missing: what are the null models and what are the hypotheses that they test? How did you shuffle the layers/edges and under which constraints? What does it mean ecologically that observed modularity was higher or lower than a particular null model? Or what does it mean for flexibility? Some of this is hidden in methods. For interlayer edges, the meaning and units are crucial. While the authors did good work in their review, it is still a bit rough. One interpretation could be that the interlayer edges represent the extent to which the dispersal is important compared to the seed dispersal within each layer. I suggest to look at studies outside ecology like Bassett et al. 2011 PNAS or others (and in particular those where Mason Porter is a co-author). Studies that used this approach would give some indication on how to interpret the interlayer coupling. In that aspect, if interlayer edges are dispersal, then it is strange that plants are connected because they don't move. So interlayer edges cannot represent dispersal in the traditional sense.

Reply: Thanks for these suggestions. Again, we agree that in the previous version the results section was too technical while much of the reasoning for the analyses was only at the end of the manuscript. We solved that issue now, so that the reader does not have to jump back and forward to understand the MS. For example, we now briefly explain how we constructed the spatial multilayer network, what the null models are testing, what is being shuffled, and the biological interpretation of the different metrics and the key results. We also provide more details about the meaning of the inter-layer coupling and included the useful literature suggestions. Some other specific questions raised in this comment are also found in several points below and we addressed them there.

L147-149: I find it a bit odd that the modularity values are above 1. The Q is normalized, or should be normalized such that Q is between 0 and 1. There may be an issue with the code, in the variable 'twomu'. Check that you add it correctly, maybe? In any case this is NOT a major issue because the value of Q per se is not relevant, but rather its value compared to the null models. Also, the module affiliation is not a function of Q . So from a science/interpretation aspect it is ok and this is a minor issue. But still, please check your code, just in case...The authors state they use the code from ref 11. But in ref 11 they use it for ordinal coupling, not categorical like in this manuscript. Maybe that is the problem?

Reply: We have double-checked all the Matlab and both the adaptation by Pilosof et al 2017 (DOI: 10.1038/s41559-017-0101) and the original by Juttla et al 2014

(<http://netwiki.amath.unc.edu/GenLouvain>) and no mistakes were found. Maximized modularity can in fact reach values above 1 (e.g. see Figure 2 in supplementary discussion in Bassett et al. 2011, DOI: 10.1073/pnas.1018985108).

L151-153: This is important but the authors don't stop to interpret this. To me, that means that as you increase the force of dispersal (or whatever it is the interlayer edges are), then the structure of the network can be explained by a random process (also depending on what the null models are). What does it mean in ecological terms, of, e.g., community structure and assembly?

Reply: Yes, the general issue of lack of ecological interpretation as already been signalled above and we are fully convinced that it is solved now. Specifically, we agree with the reviewers' interpretation and we now incorporated that view into the MS. This section now reads: "The identity of the dispersers and the intensity of movements between habitats (inter-layer strength) play a more important role for the spatial structure of the seed-dispersal network than the pattern of seed-dispersal within each individual habitat. Nonetheless, the modularity predicted by both null models tended to converge to that of the observed network at very high values of inter-layer strength (Fig. 2A; Supplementary Data 1), indicating an increasing importance of random processes in structuring the networks. This suggests that when habitat connectivity is very high the overall network structure becomes less determined by the identity of animals connecting them, and might be more contingent on the structure of seed-dispersal within habitats" (**lines 176-186**).

L162: What about module affiliation? Do the affiliations that you find make sense? Or, for example, modules contain animals from very different guilds that we would not imagine can disperse the same seeds?

Reply: We thank you for bringing this up. This type of information was indeed missing from the text. We have now clarified in the text that: "In the spatial multilayer network, modules are subsets of species that strongly interact across the different layers of the network^{27,42}. For animals, this corresponds to species that occur and disperse seeds from the same plant species in more than one habitat (Fig. 3; Fig. S2). For example, most primates (baboon, vervet monkey and *Otolemur crassicaudatus* (bush baby)) all disperse *Z. mucronata* and are consistently placed in the same module in the multilayer and in the aggregated networks, but not when habitats are weakly connected or considered independent. It is worth to note that module affiliations do not necessarily group phylogenetically related species, but species that feed on similar resources, which in seed dispersal might be mostly determined by behavioral and morphological constraints (e.g. *Corythaixoides concolor*, the go-away bird, is consistently assigned to the same module of the bush babies, Fig. 3)" (**lines 215-227**).

L167-168: That is not a sufficient definition. For example, what happens when an interaction repeats in more than one layer? do you sum them? take the average?

Reply: Yes, the aggregated network was obtained by summing all the interactions across the four habitats. We have now clarified exactly how we obtained the aggregated and the split networks (i.e. the two traditional ways in which these kind of data is explored in network literature) in order to compare the value of the multi-layer approach. These new sections have been added to the Results section: “To understand the added value of the multilayer approach in relation to the traditional monolayer approach, we compared the results from the multilayer analysis with those provided by the currently standard approaches of either merging all data into a single aggregated network ($Q_{\text{aggregated}}$), in which interactions occurring at multiple habitats are summed across habitats, or considering each habitat as a discrete and disconnected network” (**lines 187-192**) and “However, it ignored habitat connectivity because it cannot incorporate such information. In the disconnected network habitats are considered totally independently from each other, thus equivalent to calculate modularity for each of them” (**lines 196-199**) and in the Methods section: “We compared the results obtained with a multilayer network to that of two different representations of the same network: a) an aggregated network ($Q_{\text{aggregated}}$), where all interactions across the different layers were pooled to create one overall aggregated network, with the frequency of interactions that occur in multiple habitats being summed, and b) a disconnected network where habitats are considered fully independent from each other, i.e. inter-layer strength is set to zero, and thus modularity is calculated for each of them” (**lines 602-609**).

L171: This is not statistically significant!

Reply: This was in fact a mistake from us, as we meant $p \approx 0$, i.e. $p < 0.001$. Thank you for pointing this. We have now corrected this, and it now reads $p < 0.001$ as it is shown in Figure S3 (**line 195**).

L183: How do you calculate flexibility? Note that the proportion of species that change modules at least once is NOT the original definition (see Bassett 2011 PNAS). I personally do not mind the use of this term BUT this should be acknowledged and specifically defined.

Reply: Thanks for spotting this. Actually this was a mistake that has slipped from previous versions. We now replaced it for the correct term “adjustability” (rather than “flexibility”) and we explain how it was calculated and what it means the Results section: “The strength of each interaction can vary across habitats, reflecting different animal resource preferences in different contexts, and therefore, species can change their module affiliation between habitats. We calculated species adjustability as the proportion of animal or plant species that switch module affiliation at least once

between any pair of habitats” (**lines 228-232**), and in the Methods section: “For each network we calculated the mean number of modules, and the mean adjustability¹² of animal and plant species as the proportion of species in each level of the network that changed module affiliation at least once between habitats” (**lines 623-626**). We have now corrected all instances where the “flexibility”, instead of “adjustability”, occurred throughout the MS.

L183-186: Note that this is what you actually expect mathematically. So it is by definition that flexibility is negligible in high values of interlayer edges. The question here is rather quantitative. What do these particular values mean?? Why 0.7? What are the units of the interlayer edges? Or, why plants are as twice as flexible as animals? This may be difficult to interpret, but it may be worthwhile to pause and think about that.

Reply: This result is indeed what would be expected mathematically and the interpretation of the inter-layer strength units are not always straightforward. The interlayer strength can take the form of any unit that can be interpreted as an effective change between habitats (whether it's number of animals, or movement of individuals, or amount of energy). So, in this case it is not so relevant the actual value of inter-layer strength; the key aspect is that its variation has an important effect on the resulting network structure. For this reason, we removed these values from the main text and now direct the reader to the figure, where the different trends for animals and plants can be more readily perceived. Without becoming excessively technical, we have now clarified the interpretation of this result: “When the intensity of species movement between habitats (inter-layer strength) is low, animals and plants tend to interact with distinct set of species in each habitat and a higher proportion of species will change their module affiliation between habitats. As the intensity of these movements intensifies, and habitat connectivity increases, species adjustability becomes negligible and interactions tend to occur amongst the same species across all habitats. However, this stabilization on interaction partners happens at different levels of habitat connectivity for animals and plants (Fig. 3; Supplementary Fig. S4; Supplementary Data 1)” (**lines 234-242**).

L192-196: That does not give any insight into the ecology. For this, we need to know what was the null model? and also, what does it mean that species change modules across habitats? For example, in the paper by Pilosof et al 2017 NEE they interpreted this phenomenon in a temporal network in terms of different function of the species at different times. What does it mean here?

Reply: We have now added succinct information regarding the null models, and what they are testing, and also what's the meaning of changing modules/adjustability (**lines 242-257**).

L 205-207: Interaction diversity, or more properly put in ecology: "alpha diversity of interactions", is not measured by the number of interactions, but by the relative number compared to number of species, which is actually networks density (or in ecology -- connectivity). It makes sense because systems with fewer species will have less links. Please change this!

Reply: Thank you for pointing out that we used interchangeably (and incorrectly) the terms "interaction diversity" and "interaction richness", which is what we actually tested and was written in the Methods section. We have now corrected this and the sentence now reads "As for richness of interactions (...)" (**lines 264**).

L 210: this is uncommon in ecology and should be defined here, not in methods. Also the fact that it is performed on projected networks, and mention how the networks were projected.

Reply: We now provide a definition and an ecological interpretation of versatility (in the context of seed dispersal networks) in the results section. We also clarify how the network was projected: The section now reads: "We calculated each disperser multilayer versatility, which is equivalent to an overall measure of centrality to identify those that are topologically important to the structure of the spatial network. For this effect, we used a unimodal projection of the network, in which two animal species are connected if they disperse the same plant species, thus providing an insight over their likely "functional redundancy". Links between species were quantified by weighting the number of shared interactions by the assemblage size, minimizing the loss of information associated with unimodal projections. (...) The importance of these species comes from being central in the structure of the seed-dispersal network because they share plant partners with many other animals, but also because they share plant species across different habitats" (**lines 270-283**).

L 222-224: why test for this correlation? what will it teach us? why multistrength? what does it represent?

Reply: Again, thanks for highlighting the need of a deeper interpretation. The reason for this correlation was to check if the dispersers versatility (i.e. a combined measure of dispersers centrality within and across habitats) could be strongly determined by either species multistrength, specialization, or the number of habitats where it is found, in which case the metrics would be found redundant. While much of this reasoning was previously only present in the Methods section, we now improved this explanation in the Results section, that now reads: "We evaluated if the information condensed by multilayer versatility could be captured by other species-level metrics, namely specialization d' , number of habitats, and species multistrength." (**lines 290-292**). We also provide an explanation of what is multistrength, its interpretation, and the meaning of its correlation with versatility, and it now reads "Species multistrength

extends the concept of its monolayer counterpart, expressing the total number of links of a species across all layers of the network, i.e. the total shared interactions with all its neighbouring species across the habitats. However, contrary to versatility, multistrength does not account for the distribution of these links in relation to the other species, or the number of layers in which these links occurs. Thus, although both metrics are related, multistrength will not reflect the importance of a species for the overall structure of the multilayer network as much as versatility” (lines 298-305). We have also improved the interpretation of these results in the Discussion: “The relatively low correlation between multilayer versatility and multistrength, and the non-significant relationship with the number of habitats where each species occurs and its specialization (d’) reflect the information gain of using multilayer versatility, which could not be captured by conventional metrics” (lines 422-426).

DISCUSSION

L 229-230: Here also there are abrupt borders. You just connect them... I would change the focus and say that ecological networks are usually studied in a discrete form but here you connect them explicitly. This is where the novelty is. Also, would be good to down-tone the sentence (“... however convenient...”).

Reply: We followed the reviewer suggestion and down toned the opening sentences of the discussion, which now reads “Species and communities are not randomly distributed across the planet, but they are strongly structured by spatial attributes traditionally recognized by ecologists (e.g. niches, habitats, landscapes, biomes). Traditionally, species interaction networks have been studied as discrete entities with borders defined by the researchers based on different landscape attributes. However, species interactions do not abruptly finish at habitat borders, and therefore the decision of merging or segregating data from these spatial units is far from trivial” (lines 308-314).

L 231-232: This is not true. Look at Pilosof et al 2017. It is true however for spatial ecological networks.

Reply: Thanks for pointing this out. That’s what we meant, and we corrected the sentence accordingly: “inter-layer connectivity has never been explicitly incorporated in the analysis of the modular structure of spatial ecological networks” (lines 321-322).

L 258-259: This is not completely true. To be accurate -- it is the first study to explicitly incorporate the interlayer connectivity between networks in different habitats. But the authors did not really quantify this connectivity in the sense of empirical or literature-

survey data on the interlayer links. Instead, these were artificial interlayer values (whose definition and units are important here), as also mentioned 2 sentences below.

Reply: Thanks. We have now revised this sentence for greater accuracy. It now reads: “we implement for the first time a multilayer approach to evaluate the spatial structure of an ecological network explicitly incorporating the inter-layer strength connecting networks from adjacent habitats” (**lines 345-347**).

L 265-269: It is good that the authors acknowledge that but I still feel this should be a bit more profound, and more accurate: Discuss maybe how does that assumption affect the validity of the results and their interpretations? Do the results, when compared to aggregated/disconnected networks provide new insights so it was a valuable assumption to make, even though not biologically realistic?

Reply: It is quite likely that species assignment to modules would change if we had the chance to empirically measure individual inter-layer strength of all species in the field. Because we couldn't do so, we tested a whole range of possible inter-layer edge strength, so that the real (unknown) values would fall somewhere along that interval. Maybe on the previous version of the MS we were still too shy in highlighting the added value of this multilayer approach in relation to the previous aggregated or split habitats approaches. We now present a more solid case that, although aggregated and disconnected networks can correctly predict the overall network structure in some particular situations, the multilayer approach has the intrinsic advantage of not being influenced by the a priori decision of whether aggregating or disconnecting the network, and provides a more realistic depiction of the overall network, regardless of the importance of cross-habitat interactions. We have highlighted these advantages across the whole MS: Abstract: **lines 31-37**; Results: **lines 187-199** and **215-227**, where we present results for the comparison between the multilayer network and the aggregated and disconnected network; Discussion: **lines 345-396**, where we discuss the potential consequences of not having used empirically measured values for inter-layer strength but also the advantages of our approach.

L 272-273: This interpretation is not entirely accurate. It may be true if the units of the interlayer edges are number of movement events. As far as I understand it, there are two options to interpret the interlayer edges: (1) they encode the strength of interconnectivity between layers, which is correlated with movement events (more movements mean more connection); (2) they encode the relative importance of dispersal within layers to dispersal between layers. These two interpretations are clearly associated though not entirely the same. There is also the issue of connecting plants, so animal movement may not be the interpretation...The second option is already mentioned in the next sentence.

Reply: We thank the reviewer for the comment, and we agree with the reviewer. The relative strength of the intra-layer (animal-plant interaction) to the inter-layer (degree to which habitats are coupled by their common species, i.e. the intensity of the movement of the species across the habitats) that will determine the extent to which the network is modular (as it is mentioned in the sentence after). So, the structure of the spatial network is maintained even if this habitat coupling is weak, i.e. the process within habitats is stronger than the process occurring between. We have now rephrased to reflect this, and the sentence now reads “suggesting that the spatial community structure can be maintained even if the strength of the habitat connectivity is low relative to the strength of the interactions within the habitats” (**lines 371-373**).

L 280-284: This is important. It means that the animals play a similar functional role in different habitats in terms of dispersing plants. But you can only detect this if you link the habitats, even with low values of connectivity. Because if $\omega=0$ or very low then layers are independent and flexibility increases. Also, who are the animals that are flexible in values of $\omega=0.1$, for example, and what is common to them? I understand that animals in the same modules disperse similar seeds. But by looking at module affiliations does this make any sense? Or for example you find animals from very different guilds in the same module? There should be some evolutionary constraints. For example, a bird and an elephant probably will not disperse the same seeds.

Reply: Actually, such phylogenetic signal in model composition might be more common in other types of interactions where functional and morphological matching is more relevant than for seed dispersal. Seed dispersal interactions are mostly constrained by gape-size and seed/fruit size (in addition to spatial and temporal co-occurrence), and therefore, it is not surprising that we find a high diversity of animals being assigned to the same module. We have now rephrased this part of the Discussion to incorporate this information, and it now reads “Regarding the modules composition, these grouped together species that are not always phylogenetically close (e.g. primates were grouped with the go-away bird), suggesting that functional and morphological matching, such as gape-size and seed/fruit size, are more important drivers of seed-dispersal interactions. Interestingly, we detected low adjustability for most species and module affiliations remained mostly constant across habitats. Module switching occurred only for some species (e.g. primates, elephants or civets), and at very low values of habitat connectivity (Fig. 3). Most animals however, tend to disperse the same plant species in different habitats, thus maintaining a similar functional role across the landscape, even if habitat connectivity is very low. This can only be detected if the habitats are explicitly linked in the analysis of network structure” (**lines 385-396**).

L 288-289: but here flexibility was very low... how do you reconcile this with this statement?

Reply: In fact, the interpretation this result was not totally accurate, and we have now improved it (thanks for pointing out this inconsistency). The text now reads: “The capacity of species to adjust their interactions to specific contexts (thus increasing overall adjustability) is likely important for species persistence in changing environments, while at the same time tends to promote a greater connectivity (e.g. seed dispersal) across habitats. In Gorongosa, some of the species that changed module affiliation have generally wide range movements and can distribute seeds between habitats, thus giving a key contribution to plant genetic diversity and spatial distribution of plant populations through seed-dispersal” (**lines 398-406**).

L 318: Shouldn't that be intra?

Reply: Yes. We have now corrected it (**line 431**).

L 334: life history traits?

Reply: Yes, corrected as suggested (**lines 463-466**).

L 347-348: this is not a gap because it has been done before, and many times though not with networks or interlayer edges. The novelty of this study is that it explicitly links seed-seed disperser networks across habitats.

Reply: You're absolutely right, we did not want to obliterate the important legacy of all spatial ecology. We have rewritten this sentence and it now reads “Here we took a step further in the analysis of spatial mutualistic networks, and using inter-layer edges strength we explicitly considered the interactions between plants and their dispersers across multiple habitats in the analysis of the network structure” (**lines 472-475**).

METHODS

L 433: this is interesting... For plants, I am not sure that this is a good decision. It really depends on how the authors interpret the interlayer edges. Because plants do not move but they are also important for interlayer connectivity. So, if interlayer edges are dispersal, it probably does not make sense to connect plants. If interlayer edges are the degree to which layers are coupled, it may be ok to do that...

Reply: This is a very interesting point. We gave it a good thought about the issue of animal and plant movement and we now clarified our interpretation. Actually, only animals move actively across the landscape matrix, but in dispersing seeds they also move plants across habitat borders in a very tangible way (Shea 2007; How the wood moves; DOI: 10.1126/science.1136096). Therefore, and while the temporal scale

of the movements is not exactly the same, both animal and plant genes frequently cross and establish in neighbouring habitats and can be considered effective habitat connectors. As such, interlayer edges can encode the intensity of these movements of species between habitats, strengthening habitat coupling. We have clarified this point in **lines 581-587**.

L 452: It is actually calculated using the function Q. General Louvain is a search function to find that maximum Q.

Reply: Now corrected to: “The Q modularity function was maximized applying a “generalized Louvain” method” (**lines 602-604**).

L 503: which was?

Reply: This reduced model was an intercept-only model (as “habitat” is the only explanatory variable). It now reads: “against a reduced (i.e. intercept-only) model” (**line 665**). We also clarified this issue in the legend of Supplementary Table S3, where AIC values are reported.

L 514-515: Actually, projected networks depict the niche overlap between pairs of animals (Mello, 2015. *Oikos* 124:1031–1039.). So central animals are those that share seeds with many other animals, and versatile animals are those that also share many seeds across habitats

Reply: You’re absolutely right. Please see our joint reply after the next comment, which is very much related.

L 515-516: But how is that done? what are the values of the edges? the total number of plants shared? a measure of b-diversity between pairs of animals? binary?

Reply: Indeed, our original explanation was lacking some important details about the unimodal projections. We now added that needed explanation and indicate how links weight was estimated: “The implementation of this method requires bipartite networks to be projected onto unimodal networks. While some projection methods entail some loss of information, we applied a weighted projection which estimates interaction weight based on the proportion of shared interactions (i.e. seed species shared by disperser species) relatively to the total network size, thus minimizing the loss of information. The projection was performed with function `projecting_tm` from the R package `tnet`. This algorithm is particularly suitable to multilayer networks as it condensates information on dispersers niche overlap⁴³, based on the importance of their shared dispersed seeds” (**lines 675-684**).

Fig 1: Would be good to add how the aggregated network was obtained

Reply: We have now added this information to the legend of Fig 1 “The aggregated network was obtained by pooling all interactions across the different habitats, and frequencies being the sum of its frequencies in all habitats”.

Typos and grammar corrections

L75: deleted “sciences”, and replaced by “research areas” (**line 81-82**).

L 216: deleted “however the importance”, and replaced by “The importance” (**line 280**).

L 218: deleted “Overall”, and replaced by “across different habitats” (**line 282-283**).

L 219: deleted “over”, and replaced by “more than” (**line 288**).

L 239: “understanding” instead of “understand” (**line 327**).

L 253-254: removed repetition of “for the”, and replaced “dynamic” by “dynamics” (**line 340-341**).

L 265: “empirically” instead of “empirical” (**line 355**).

L 293: “structured at a much finer scales” instead of “structure at much finer scale” (**line 408**).

L 346-347: We have rewritten this and it now reads: “a multilayer approach is a most valuable tool to explore these factors” (**line 471-472**).

L 359: replaced “study” with “link” (**line 485**).

L 363: replaced “will likely” with “may” (**line 487**).

L 364: replaced “misidentify” with “misidentification of” (**line 489-490**).

Reviewer #3 (Remarks to the Author):

The authors have significantly revised the manuscript. My main concern on the original version was that the questions and the applicability of results beyond this specific system were unclear, and did not do justice to the detailed dataset. I think this aspect has been much improved, with specific questions being clarified, previous work on cross-habitat linkages being better recognised, and a greater focus on concepts rather than methodology in the introduction.

I think though, that this focus could be clearer in the abstract (e.g. L.30-33). Much of this will mean nothing to readers unless they read the whole paper (e.g. the abstract doesn't even say what the layers are, so interlayer connectivity is meaningless). I'd focus the abstract more around the main questions of the paper (L. 110-124): i.e. how do multilayer networks with spatial connections among habitats (i.e. layers) differ from either the single habitats or the habitats just aggregated together? Also, what does a disperser's versatility tell you that traditional species-level metrics don't? I'd give precise answers to these questions in the abstract.

Reply: We fully agree with this comment and the basically rewritten the abstract, which is now much more accessible, informative and focused on the main advances of this work.

- I am left with only one major concern about the ms. L.418 Interaction frequencies were calculated by pooling together data that were sampled in very different ways, and I am concerned that this could generate some biases. For example, the probability of observing dung along a transect will correlate with the size of the dung, and I expect that this was the reason that the authors used different approaches (mist netting etc) to identify dispersal by birds. However, because the analysis compares species with each other, it is absolutely necessary that sampling is equivalent across species. Otherwise, some species could appear to be less important as connectors simply because they are under-sampled. I'm not sure what the retention time of dung would be (before decomposition or removal by insects), but I imagine that dung could persist over several days (or even weeks), and will be more likely to do so if it is large (i.e. comes from a large animal). This means that transects would capture a large number of dispersal events by even a single individual large mammal, whereas mist nets only capture a single fecal sample from each bird. Consequently, I would expect interactions by birds to be undersampled relative to interactions by animals (this suspicion was confirmed on L.127), and therefore I don't believe that the interaction frequencies are comparable (i.e. on the same scale) for species sampled using different techniques (birds vs. mammals). For the mixed models that conduct analyses within species, it would appear that this doesn't matter. However, if some plant species have a bird-dispersed strategy, they will appear to be less important and birds will appear to be less central than they should be (L.211-212 confirms that the species with highest versatility are mammals, and the conclusion on L.242-246 could be an artefact of many species' interlayer connectivity being poorly sampled). I could see two potential solutions to this problem: 1) the networks could be treated as unweighted, or 2) the

sampling completeness analysis could be conducted separately for each method of sampling, to demonstrate that interactions (in addition to species) were sampled equivalently using the different methods, such that the frequencies captured by each method are directly comparable as link weights within a single network.

Reply: Although still not routinely considered in network studies, the quality of the sampling completeness is of course essential to derive meaningful network descriptors, and for that reason an issue in which we are particularly interested as a research group (see Costa et al. (2016) Sampling completeness in seed dispersal networks; DOI:10.1016/j.baae.2015.09.008). Unfortunately, as most networks are the result of a single sampling protocol/method, the potential consequences of merging data obtained from different sampling protocols into a single interaction matrix remain unexplored. We are actually planning to evaluate this issue with a different dataset. Nevertheless, the main potential bias here would be if some guilds, explored by a particular sampling method were much better sampled than others. In order to test this, we have now included in the MS an estimation of the sampling completeness attained with each sampling method (**lines 444-445**). We find that interaction completeness is consistently low for all sampling methods (Min 13%. - Max 25%), and therefore the data could be analysed together. Actually, this analysis shows that although we analysed more mammal droppings (mostly collected along transects) than bird droppings (mostly collected with mist nets), only a very small proportion of the bird droppings analysed (8%) contained seeds, compared to a much greater occurrence of seeds in mammal dung (34%). This shows that the relative greater versatility of mammals actually real and not an artefact due to a potentially lower sampling intensity of mist-netting. While our objective was not to achieve 100% of sampling completeness with each method, but only to show that their results are comparable, it is important to note that these estimations should be taken only as a relative indication as they are strongly underestimated. Actual sampling completeness is actually much greater than predicted by species accumulation curves, first because these populations (sampled during an entire year) are not closed populations – due to advance phenology throughout the season that brings new species into contact; and secondly due to the existence of many structural forbidden links, i.e. potentially predicted interactions that cannot occur due to temporal, spatial, and physiological mismatching between species (see Jordano 2016 Sampling networks of ecological interactions, DOI: 10.1111/1365-2435.12763). Moreover, it has been recently demonstrated that merging data from distinct, sampling methods to assemble interaction matrices improves the quality of the data due to the complementarity of methodologies (Escribano-Ávila, G., et al. (in press), Seed dispersal networks in the tropic. Chapter in book "Ecological Networks in the Tropics" edited by W. Dattilo & V. Rico-Gray, Springer). As for long-distance seed dispersal strategies, there isn't a strong deterministic effect of the dispersal syndrome and the actual dispersal vector/guild (Higgins et al 2003, Ecology DOI:10.1890/01-0616; Heleno & Vargas 2015 DOI:10.1111/geb.12273), which suggests that sampling method should not be highly related with the probability that each plant species will be found most important.

For all of the above, we decided to follow the recommendation of review#3 and include more data on sampling completeness (specifically regarding interaction sampling completeness achieved with each method), and added a specific caveat to address this issue in the discussion (**lines 439-460**). We considered that conducting an alternative analysis on rarefied data would create other equally important sources of bias and we would risk “throwing the baby out with the bathwater”.

My remaining comments are all very minor:

L.24 “and static” may be true in the majority of cases, but there are still quite a large number of exceptions to this, so I’d delete it.

Reply: Corrected as suggested (**lines 23-24**).

L.70, I think the names of the authors for ref 14 are missing (or at least some noun is needed where the ref number is).

Reply: Corrected as suggested (**line 76**).

L.84 grammar “nodes that can change module”. Perhaps modules plural, and change how? Or do you mean that can occupy different modules in different layers?

Reply: This sentence was not very clear, indeed. We meant nodes that can belong to different modules in different layers. We changed it to make the sentence clearer, and it now reads (**lines 91**).

L.91 change to “considerably”.

Reply: Corrected (**line 99**).

- The paragraph beginning L.90 doesn’t work very well. First it says that species’ roles are conserved, and then that centrality is often used to measure species roles. This implies that the references for role conservatism in the first sentence used centrality, and that’s not the case (at least not in the Stouffer et al one, with which I’m most familiar). So I think the justification for focusing on centrality needs to be better defended. I’m not even sure it’s needed here, given that only modularity is mentioned in the specific questions below.

Reply: We agree that the paragraph could be misleading. We clarified what we meant, and added a more appropriate reference: Emer et. al. (2016), DOI: 10.1111/ddi.12458).” (**line 98**).

L.119, L.199 “disperser” singular for adjectives

Reply: Corrected (**lines 123 and 259**).

L.167-9 “The structure...being significantly lower” How can structure be lower? Do you mean modularity?

Reply: Indeed, it is modularity that is lower. We have now corrected this and it now reads “with modularity being significantly lower than” (**line 194**).

L.172 add “, which” after modules

Reply: Corrected (**line 210**).

L.229 “habitat borders”

Reply: Corrected (**lines 313**).

L.238 “understanding”

Reply: Corrected (**line 327**).

L.246 spelling “ensuring”

Reply: Corrected (**line 334**).

L.252-3, words repeat

Reply: Corrected (**line 341**).

L.261-263 Doesn't this contradict the result on L.172-173 that the aggregated network was in line with the multilayer network? I would explain this result in laypersons terms, as it's the first main objective of this paper to compare the two approaches. Something along the lines of “This means that species may appear to interact with others in different parts of the network (i.e. across modules), but they tend to do this only across certain habitats, such that species tend to interact with subsets of species within subsets of spatially-coupled habitats”. This probably isn't exactly right, but an explanation in simple terms like this would be useful.

Reply: Thank you for spotting this apparent contradiction. We have revised these two sections, using much of your suggested text, to clarify that although some metrics provide similar results (e.g. number of modules), the structure revealed by the two approaches is not the same and multilayers provide new insights. It now reads: “Our spatial multilayer seed-dispersal network exhibited a highly modular structure, i.e. species tend to interact with subsets of species (i.e. modules) within subsets of spatially-coupled habitats. By explicitly including non-zero inter-layer links, i.e. the habitat connectivity promoted by the common species, it is possible to account for the inter-dependence of the network structure across multiple habitats, and identify modules that spread across habitat borders” (**lines 347-353**), and “Importantly, the structure of the seed-dispersal network was not fully captured by the aggregated network or by considering each habitat as an independent network. Consequently, the result obtained by using a multilayer approach is not biased by any decision regarding aggregating, or disconnecting the different layers of the network. Instead, the resulting structure is a consequence of the relative importance between the processes occurring within and between layers, which is objectively defined by the relative strength of the inter- to intra-layer edges” (**lines 377-384**).

L.264 “empirically”

Reply: Corrected (**line 355**).

L.279 across what? (noun missing).

Reply: We meant “processes spanning across different layers”. It is now corrected (**line 368-369**).

L.403 change “was” to “were”

Reply: Corrected (**line 530**).

L.468-469 The interlayer strength was the same for all species. I take it that this was the same for all species that actually had an interlayer link (i.e. occurred in two habitats), not all species in the aggregated network? Perhaps add a couple of words to clarify this

Reply: Yes, absolutely. We clarified this point and the sentence now reads “i.e. all species connecting any two pair of habitats have the same effect in the inter-layer process” (**line 598-600**).

L.434 The authors use occurrence in multiple habitats as a proxy for movement. The caveat should be made that occurrence across habitats doesn't equal frequent dispersal across habitats, which is what actually matters for cross-habitat connectivity. I realise you don't have another option here, but it's still an unproven assumption.

Reply: Thank you for your comment. This is in fact in a similar vein to a comment from reviewer #2, regarding the ecological interpretation inter-layer links, which was indeed not very clear in the previous version. We have now clarified this interpretation in several sections through the MS (**lines 157-161** in the Results, **354-369** in the Discussion, and **560-566** in the Methods). These sections now clarify that inter-layer edges should ideally reflect actual movement of individuals (or other form of matter or energy) between the different habitats. Unfortunately that data is rarely available for all species of a community, and instead we used co-occurrence of the same species in two habitats to infer a potential movement/connection between the two. We clarified that this is only a sub-optimal working proxy of real movement, in the same way that the abundance of species across successive temporal windows can be used as a proxy for the transition (i.e. survival) of individuals across time.

L.474 "layers" plural

Reply: Corrected (**line 605**).

L. 497 The use of d' as a specialization measure is increasingly criticised, in part because of its low informativity (signal to noise ratio, see Poisot et al. 2012), and in part because it measures selectivity within a range of resources, rather than the range itself, so its interpretation can be counterintuitive when compared with other measures like degree. If you're going to use only this metric, it should be defended.

Reply: We agree with the limitations that have been pointed to this metric, although we also see some important advantages of using d' . We now provide the reader with a very brief outline of this criticisms and the rationale to use d' in this work: "First we assessed whether the specialization of seed dispersers differed consistently between habitats by calculating animal specialization (d'), which quantifies their selectiveness for seeds within the range of resources used⁷⁹. However, the number of interactions of a species is considered to reflect both resources availability and consumer activity. This metric takes into account the pattern of interaction of a species in relation to the available resources, while being robust to sampling effort, network size and asymmetry (Blughthen et al 2006, DOI: 10.1186/1472-6785-6-9, but see Poisot et al. 2012, DOI: 10.1111/j.2041-210X.2011.00174.x)" (**lines 633-639**).

L.502 spelling "Akaike"

Reply: Corrected (**line 644**).

L.505 “Residuals were inspected for departures from normality”. Why? I thought you were assuming gamma, not normal errors (L. 498).

Reply: Yes, we are modelling the response against a gamma distribution and therefore we deleted the sentence to avoid potential confusions.

L.547 I think nowadays “data available upon request” isn’t enough. They should be made available in a repository (or provided as supplementary material since this journal isn’t paywalled).

Reply: This issue has been discussed with the editor and data will be made available as soon as possible, either upon request or on open repositories, unfortunately we cannot do it at the time of publication.

L.564 “led”, not “lead”

Reply: Corrected (**line 706**).